# EFFICIENT TAIL-AWARE GENERATIVE OPTIMIZATION VIA FLOW MODEL FINE-TUNING

**Zifan Wang**
KTH
zifanw@kth.se

**Riccardo De Santi**
ETH Zurich, ETH AI Center
rdesanti@ethz.ch

**Xiaoyu Mo**
KTH
xmo@kth.se

**Michael M. Zavlanos**
Duke University
michael.zavlanos@duke.edu

**Andreas Krause**
ETH Zurich, ETH AI Center
krausea@ethz.ch

**Karl H. Johansson**
KTH
kallej@kth.se

## ABSTRACT

Fine-tuning pre-trained diffusion and flow models to optimize downstream utilities is central to real-world deployment. Existing entropy-regularized methods primarily maximize expected reward, providing no mechanism to shape tail behavior. However, tail control is often essential: the lower tail determines reliability by limiting low-reward failures, while the upper tail enables discovery by prioritizing rare, high-reward outcomes. In this work, we present **T**ail-aware **F**low **F**ine-**T**uning (TFFT), a principled and efficient distributional fine-tuning algorithm based on the Conditional Value-at-Risk (CVaR). We address two distinct tail-shaping goals: right-CVaR for seeking novel samples in the high-reward tail and left-CVaR for controlling worst-case samples in the low-reward tail. Unlike prior approaches that rely on non-linear optimization, we leverage the variational dual formulation of CVaR to decompose it into a decoupled two-stage procedure: a lightweight one-dimensional threshold optimization step, and a single entropy-regularized fine-tuning process via a specific pseudo-reward. This decomposition achieves CVaR fine-tuning efficiently with computational cost comparable to standard expected fine-tuning methods. We demonstrate the effectiveness of TFFT across illustrative experiments, high-dimensional text-to-image generation, and molecular design.

## 1 INTRODUCTION

Flow Lipman et al. (2022); Liu et al. (2022) and diffusion models Ho et al. (2020); Song and Ermon (2019); Song et al. (2020) have emerged as dominant paradigms in the rapid advancement of large-scale generative modeling. Their capacity to generate high-fidelity samples has enabled widespread adoption in fields such as chemistry Hoogeboom et al. (2022), biology Corso et al. (2022), and robotics Chi et al. (2025). While learning to approximate the data distribution has enabled impressive generative models, it is often insufficient for real-world tasks such as scientific discovery (Bilodeau et al., 2022; Zeni et al., 2023). In these settings, one typically seeks samples that optimize task-specific utilities. A common approach is to fine-tune the generative model to maximize the expected reward of generated samples, while preserving information from the pre-trained model via KL-divergence regularization. Importantly, this fine-tuning objective admits an interpretation as an entropy-regularized control problem (e.g., Domingo-Enrich et al., 2024; Uehara et al., 2024a; Tang and Zhou, 2024), enabling practical and effective procedures with demonstrated success in image generation (Domingo-Enrich et al., 2024), molecular design (Uehara et al., 2024b), and protein engineering (Uehara et al., 2024b).

Unfortunately, many practically relevant objectives fall outside this mean-reward formulation. As noted by De Santi et al. (2025), optimizing the expectation is inadequate when utility is governed by the tail behavior of the reward distribution. Examples are *risk-averse* and *novelty-seeking* reward maximization. In risk-averse settings, e.g., robust image synthesis Schramowski et al. (2023), one seeks to suppress lower-tail failures by steering the model toward distributions with improved worst-case rewards, thereby enhancing reliability and safety. Similarly, in novelty-seeking settings, e.g., de novo molecular design Jain et al. (2022); Dunn and Koes (2024), one typically wishes

to sacrifice the average reward of generated designs toward increasing the chances of generating exceptionally high-reward samples. We denote this class of problems by tail-aware generative optimization (De Santi et al., 2025).

To address the limitation of expected reward maximization, De Santi et al. (2025) recently proposed Flow Density Control (FDC), the first framework to integrate Conditional Value-at-Risk (CVaR) constraints into generative optimization. CVaR is a coherent risk measure Rockafellar et al. (2000) that evaluates the expectation of a distribution's tail, allowing for rigorous control over extreme outcomes. Specifically, this allows for two distinct objectives: right-CVaR, the expected reward conditioned on the upper tail, which promotes novelty-seeking; and left-CVaR, the expected reward conditioned on the lower tail, which captures risk aversion. Although FDC demonstrated the ability to perform such tail-aware generative optimization, it tackles the inherent non-linear optimization problem via sequential use of an entropy-regularized fine-tuning scheme. On the contrary, in this work we propose **T**ail-aware **F**low **F**ine-**T**uning (TFFT), a method that solves both risk-averse and novelty-seeking generative optimization problems via a single use of any entropy-regularized fine-tuning method. This renders TFFT significantly more computationally efficient.

A fundamental contribution of this work lies in leveraging the variational dual formulation of CVaR to demonstrate that the optimization decouples into a highly efficient two-stage procedure. The first stage involves a lightweight one-dimensional optimization problem to determines the risk-sensitive quantile threshold using the pre-trained generative model. The second stage is a single standard entropy-regularized fine-tuning routine using a pseudo-reward based on that optimal threshold. This decomposition is significant as it renders risk-averse and novelty-seeking fine-tuning practically solvable with computational cost comparable to standard expected reward maximization, avoiding the need to use sequential fine-tuning solvers. We empirically validate TFFT across a hierarchy of tasks. First, we consider illustrative settings to visually assess the method's precise control over tail behaviour. Then, we utilize risk-averse reward optimization to optimize the worst-case performance in a text-to-image generation task, thereby minimizing the occurrence of low-quality samples, and evaluate TFFT novelty-seeking reward maximization for molecular design to prioritize high-rewards candidates.

## 2 Preliminaries and Notation

### 2.1 Generative Flow and Diffusion Models

Flow-based generative models aim to approximately sample data points from a data distribution $p_{\text{data}}$ Lipman et al. (2024). A convenient perspective is to construct a continuous-time transport that maps samples from a simple source distribution $p_0$ (e.g., a standard Gaussian) into samples from $p_{\text{data}}$.

A generative flow is a time-dependent map $\psi : [0,1] \times \mathbb{R}^d \to \mathbb{R}^d$, $(t,x) \mapsto \psi_t(x)$, which transports an initial random variable $X_0 \sim p_0$ along the trajectory $X_t = \psi_t(X_0)$, $t \in [0,1]$, so that ideally $X_1 \sim p_{\text{data}}$. A common parametrization defines $\psi_t$ through a time-dependent velocity field $u : [0,1] \times \mathbb{R}^d \to \mathbb{R}^d$, via the flow ODE

$$\frac{d}{dt}\psi_t(x) = u_t(\psi_t(x)), \qquad \psi_0(x) = x. \tag{1}$$

In other words, the velocity field $u_t(\cdot)$ specifies an infinitesimal displacement rule that deterministically transports points over time. Let $p_t$ denote the marginal density of $X_t$. The pair $(u, \{p_t\}_{t \in [0,1]})$ is consistent with the flow dynamics if the induced density path satisfies the continuity equation

$$\partial_t p_t(x) + \nabla \cdot \big(p_t(x)\, u_t(x)\big) = 0,$$

which expresses conservation of probability mass under the deterministic transport. We write $p^u := \{p_t^u\}_{t \in [0,1]}$ for the marginal density path induced by a velocity field $u$.

Standard training techniques (e.g., flow matching) learn a parameterized velocity field $u^\pi$ such that the induced path satisfies $p_1^{u^\pi} \approx p_{\text{data}}$. After training, sampling proceeds by drawing $X_0 \sim p_0$ and numerically integrating the ODE equation 1 to obtain $X_1 \sim p_1^{u^\pi}$. Although diffusion models (Song and Ermon, 2019) are typically introduced through stochastic dynamics, they can be re-expressed via an ODE preserving the same marginal densities (Lipman et al., 2024, Chapter 10). Therefore, our results apply to diffusion models as well.

## 2.2 Entropy-Regularized Flow Fine-Tuning

Let a generative flow model be parameterized by $\pi$, and we denote the induced sampling distribution $p_1^{u^\pi}$ by $p^\pi$ for simplicity of notation. Assume that the model is pre-trained, and can generate samples from the distribution $p^{\mathrm{pre}}$. Given a reward function $r : \mathcal{X} \to \mathbb{R}$ that scores samples, the goal of fine-tuning is to learn a new sampling distribution that increases the reward while remaining close to $p^{\mathrm{pre}}$.

A widely used objective is entropy-regularized reward maximization, also referred to as linear generative optimization

$$\max_{p^\pi \in \mathcal{P}} \mathbb{E}_{x \sim p^\pi}[\, r(x)\,] - \alpha\, D_{\mathrm{KL}}(p^\pi \,\|\, p^{\mathrm{pre}})\,, \tag{2}$$

which optimizes the expectation of the reward distribution, penalized by a KL divergence term multiplied by $\alpha$. Throughout the paper, we assume that the set of admissible policies induces a space of probability distributions $\mathcal{P} = \{p^\pi\}$ that is convex and compact. We also assume that the reward distribution induced by the pre-trained model $p^{\mathrm{pre}}$ is continuous (i.e., admits a density with no atoms). The solution of equation 2 admits the closed-form Boltzmann distribution $p^\star(x) \propto p_0(x) \exp\left(\frac{r(x)}{\alpha}\right)$, and can be solved via entropy-regularized control methods Uehara et al. (2024a); Domingo-Enrich et al. (2024).

## 3 Problem Statement: Tail-Aware Generative Optimization via Flow Model Fine-Tuning

The conventional entropy-regularized fine-tuning objective equation 2 maximizes the expected reward under the learned distribution and therefore does not directly capture tail-sensitive goals. In discovery settings such as molecular design or creative generation, the primary aim is often to increase the probability of producing a small number of exceptional candidates; this can require trading off average performance in order to allocate more mass to the upper tail. Conversely, in safety-critical applications where avoiding low-reward failures is central, an expected-reward objective provides too weak an incentive to improve the lower tail, which captures worst-case outcomes. These considerations motivate the following CVaR-based objectives (De Santi et al., 2025).

Let $\beta \in (0,1)$ be a risk level and $\alpha > 0$ be a regularization coefficient. We consider the CVaR fine-tuning problem

$$\max_\pi \mathcal{G}(\pi) := \mathrm{CVaR}_{\beta,p^\pi}[r(X)] - \alpha D_{\mathrm{KL}}(p^\pi \| p^{\mathrm{pre}}), \tag{3}$$

where CVaR is a risk measure that evaluates the expected value of the tail of a distribution. To define it rigorously, let $X \sim p$ denote the random variable governed by the generative policy, and let $Z = r(X)$ be the corresponding scalar reward variable. The Value-at-Risk (VaR) at level $\beta$ is defined as the $\beta$-quantile of the reward distribution $\mathrm{VaR}_\beta[p] = \inf\{\zeta : \mathbb{P}_{X \sim p}(r(X) \leq \zeta) \geq \beta\}$. We consider two types of CVaR: right-CVaR defined as $\mathrm{R\text{-}CVaR}_{\beta,p}[r(X)] = \mathbb{E}_{X \sim p}[r(X) \mid r(X) \geq \mathrm{VaR}_\beta[p]]$ and left-CVaR defined as $\mathrm{L\text{-}CVaR}_{\beta,p}[r(X)] = \mathbb{E}_{X \sim p}[r(X) \mid r(X) \leq \mathrm{VaR}_\beta[p]]$. Their variational dual formulations Rockafellar et al. (2000) are:

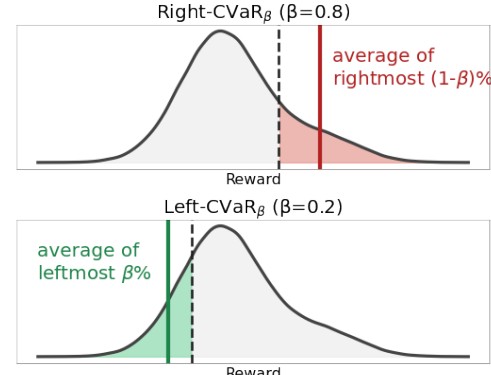

Figure 1: Illustration of right-CVaR and left-CVaR.

$$\mathrm{R\text{-}CVaR}_{\beta,p}[r(X)] = \min_{t \in \mathbb{R}} \left\{ t + \frac{1}{1-\beta} \mathbb{E}_{x \sim p}[r(x) - t]_+ \right\},$$

$$\mathrm{L\text{-}CVaR}_{\beta,p}[r(X)] = \max_{t \in \mathbb{R}} \left\{ t - \frac{1}{\beta} \mathbb{E}_{x \sim p}[t - r(x)]_+ \right\},$$

where $[z]_+ := \max(0, z)$. An illustration of right- and left-CVaR is shown in Fig. 1. Crucially, the CVaR in equation 3 can be either right-CVaR for novelty-seeking or left-CVaR for risk-aversion.

## 4 VARIATIONAL DUAL FORMULATION OF CVaR FOR EFFICIENT TAIL-AWARE FLOW FINE-TUNING

The CVaR fine-tuning objective defined in equation 3 is non-linear with respect to the probability distribution $p^\pi$. Unfortunately, this non-linearity precludes the direct application of standard entropy-regularized control methods. Flow density control (FDC) De Santi et al. (2025), a general framework for non-linear generative optimization (GO), addresses this difficulty by decomposing the nonlinear GO problem into a sequence of $K$ linear GO sub-problems, each requiring a full run of a fine-tuning solver, e.g., Adjoint Matching (Domingo-Enrich et al., 2024). While this approach is universal, the computational cost scales linearly with $K$, making it significantly more computationally expensive than standard entropy-regularized control methods. In the following, we demonstrate that the specific variational structure of CVaR allows us to bypass this iterative procedure entirely. This is achieved through the reformulations of equation 3, which we provide below for both right- and left-CVaR. Detailed proofs are provided in Appendix B.1 and Appendix B.2.

**Theorem 4.1** *Let $\beta \in (0, 1)$ and $\alpha > 0$. Assume that the reward $r(x)$ is bounded. Then, the right-CVaR fine-tuning problem admits the equivalent reformulation:*

$$\max_{p^\pi} \ \text{R} - \text{CVaR}_{\beta, p^\pi}[r(X)] - \alpha D_{\text{KL}}(p^\pi \parallel p^{\text{pre}}) \tag{4}$$

$$= \min_{t \in \mathbb{R}} \left\{ t + \alpha \log \mathbb{E}_{X \sim p^{\text{pre}}} \left[ \exp \left( \frac{[r(X) - t]_+}{\alpha(1 - \beta)} \right) \right] \right\}. \tag{5}$$

*Moreover, let $t^*$ be the optimal threshold minimizing equation 5. The optimal distribution maximizing equation 4 is given by*

$$p_R^*(x) \propto p^{pre}(x) \exp \left( \frac{[r(X) - t^*]_+}{\alpha(1 - \beta)} \right). \tag{6}$$

*Finally, it satisfies that $\text{VaR}_\beta(p_R^\star) = t^\star$.*

**Theorem 4.2** *Let $\beta \in (0, 1)$ and $\alpha > 0$. Assume that the reward $r(x)$ is bounded. Then, the left-CVaR fine-tuning problem admits the equivalent reformulation:*

$$\max_{p^\pi} \ \text{L} - \text{CVaR}_{\beta, p^\pi}[r(X)] - \alpha D_{\text{KL}}(p^\pi \parallel p^{\text{pre}}) \tag{7}$$

$$= \max_{t \in \mathbb{R}} \left\{ t + \alpha \log \mathbb{E}_{X \sim p^{\text{pre}}} \left[ \exp \left( -\frac{[t - r(X)]_+}{\alpha\beta} \right) \right] \right\}. \tag{8}$$

*Moreover, let $t^*$ be the optimal threshold minimizing equation 8. The optimal distribution maximizing equation 7 is given by*

$$p_L^*(x) \propto p^{pre}(x) \exp \left( -\frac{[t^* - r(X)]_+}{\alpha\beta} \right). \tag{9}$$

*Finally, it satisfies that $\text{VaR}_\beta(p_L^\star) = t^\star$.*

Theorems 4.1 and 4.2 reveal that the non-linear CVaR GO problem effectively decomposes into two tractable sub-problems. First, one solves a one-dimensional scalar optimization over the threshold, using only samples from the pre-trained distribution $p^{\text{pre}}$. Second, conditioning on this threshold, the optimal fine-tuned distribution is characterized as the solution to a standard entropy-regularized fine-tuning problem with a specific pseudo-reward function.

## 5 ALGORITHM: TAIL-AWARE FLOW FINE-TUNING

Guided by the reformulations established in Sec. 4, we now introduce **T**ail-aware **F**low **F**ine-**T**uning (TFFT) (see Alg. 1) by describing in the following its two main stages.

**Stage 1: Efficient Threshold Optimization.** The primary goal of this stage is to identify the threshold that delineates the tail region of interest. We define the scalar objectives

$$F_R(t) := t + \alpha \log \mathbb{E}_{x \sim p^{\text{pre}}} \left[ \exp \left( \frac{[r(x) - t]_+}{\alpha(1 - \beta)} \right) \right], \tag{10}$$

$$F_L(t) := t + \alpha \log \mathbb{E}_{x \sim p^{\text{pre}}} \left[ \exp \left( -\frac{[t - r(x)]_+}{\alpha\beta} \right) \right], \tag{11}$$

---

**Algorithm 1** TFFT: Tail-aware Flow Fine-Tuning

---

1: **Input:** Pre-trained model $u^{\text{pre}}$, reward $r(x)$, risk level $\beta$, mode $\in \{\text{Right}, \text{Left}\}$.
2: // *Stage 1: Optimal Threshold Optimization*
3: **if** mode = Right **then**
4:     $t^* = \arg\min_t F_R(t)$.
5: **else**
6:     $t^* = \arg\max_t F_L(t)$.
7: **end if**
8: // *Stage 2: Single Fine-Tuning Step*
9: **if** mode = Right **then**
10:     Construct pseudo-reward: $r^*(x) = \frac{[r(x)-t^*]_+}{1-\beta}$.
11: **else**
12:     Construct pseudo-reward: $r^*(x) = -\frac{[t^*-r(x)]_+}{\beta}$.
13: **end if**
14: $u^{\text{finetune}} \leftarrow \text{FineTune}(p^{\text{pre}}, r^*)$.                      $\triangleright$ via Adjoint Matching
15: **Output** $u^{\text{finetune}}$

---

for right- and left-CVaR, respectively. The global optimization of these objectives is tractable due to their favorable theoretical properties as we provide below.

**Theorem 5.1** $F_R$ *is strictly convex and smooth with Lipschitz constant* $L_R = \frac{1}{4\alpha(1-\beta)^2}$. *Conversely,* $F_L$ *is strictly concave and smooth with Lipschitz constant* $L_L = \frac{1}{4\alpha\beta^2}$.

This theorem ensures that the optimization landscape is unimodal, meaning any stationary point is the unique global optimum. Consequently, $t^*$ can be reliably determined using gradient-based methods. In practice, we approximate these expectations using a finite batch of $N$ samples $\{x_i\}_{i=1}^N$. Crucially, these samples are drawn from the pre-trained prior $p^{\text{pre}}$, which allows the samples and their reward scores to be collected offline and reused. Ultimately, the combination of favorable theoretical properties, the sufficiency of offline samples, and the low dimensionality ensures that this stage is highly efficient. The resulting computational overhead is negligible compared to the subsequent generative fine-tuning.

**Stage 2: Threshold-Conditioned Fine-Tuning.** Once $t^*$ is identified, TFFT constructs a *pseudo-reward* $r^*(x)$ to guide fine-tuning. For right-CVaR, we define $r^*(x) = \frac{[r(x)-t^*]_+}{1-\beta}$, which creates non-zero gradients solely for novel samples exceeding the threshold. Conversely, for left-CVaR, we define $r^*(x) = -\frac{[t^*-r(x)]_+}{\beta}$, which penalizes only the worse failures below the threshold. The constructed pseudo-reward is then used by a standard entropy-regularized solver, such as Adjoint Matching (Domingo-Enrich et al., 2024), to update the policy $\pi$ in a single fine-tuning step.

### 5.1 COMPARISON TO NON-LINEAR GENERATIVE OPTIMIZATION

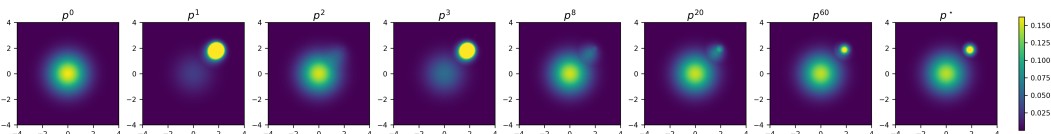

Figure 2: Evolution of distributions $p^k$ under the FDC update rule equation 20 in the 2D example. The sequence $p^k$ converges to our characterized target distribution $p_R^*$ defined in equation 6, which empirically validates our theory and confirms that TFFT bypasses the expensive iterative loop.

While TFFT leverages the variational dual form of CVaR to yield a two-stage procedure, an alternative and widely used route for handling non-linear objectives is to iteratively linearize them, reducing optimization to a sequence of tractable subproblems. FDC (De Santi et al., 2025) brings this idea to GO over the infinite-dimensional space of probability distributions (visualized in Fig. 2). A comparison of computational complexity is presented in Table 1, highlighting TFFT's clear efficiency advantage over FDC. More details are provided in Appendix D.

Table 1: Generative optimization computational complexity comparison. $C_{\text{FT}}$ denotes the cost of a single entropy-regularized fine-tuning run. $C_{\text{Thresh}}$ denotes the cost of the scalar threshold optimization in Stage 1. $K$ denotes the number of iterations required for general functional optimization.

|        | Utility | Fine-tuning Calls | Complexity | Exactness |
|--------|---------|-------------------|------------|-----------|
| Method | General | $K$ | $\mathcal{O}(K \cdot C_{\text{FT}})$ | no |
| FDC | CVaR | 1 | $\mathcal{O}(C_{\text{Thresh}} + C_{\text{FT}})$ | yes |

## 6 THEORETICAL ANALYSIS

We provide a theoretical guarantee for the two-stage procedure of TFFT. We first establish that finding $t^*$ in Stage 1 is tractable, and then show that the Stage 2 target distribution is robust to small numerical errors in $t^*$.

### 6.1 ANALYSIS OF STAGE 1

The threshold optimization in Stage 1 is a one-dimensional problem. As established in Theorem 5.1, the objective functions $F_R(t)$ (convex) and $F_L(t)$ (concave) are $L$-smooth, ensuring a well-behaved optimization landscape.

A practical challenge arises from the stochastic estimation of the gradients. Due to the log-expectation structure of the objective, the derivatives $F'_R(t)$ and $F'_L(t)$ inherently take the form of a *ratio of expectations*. When we approximate these gradients using a finite batch of $N$ samples, we encounter a fundamental statistical issue: the expectation of a ratio is not the ratio of expectations (i.e., $\mathbb{E}[\hat{A}/\hat{B}] \neq \mathbb{E}[\hat{A}]/\mathbb{E}[\hat{B}]$). Consequently, the gradient estimators are biased.

Despite this bias, we establish that the optimization remains tractable. Under the mild assumption of bounded rewards, standard concentration arguments ensure that both the bias and the variance of the estimators decay at a rate of $\mathcal{O}(1/N)$. This property allows us to guarantee convergence to a neighborhood of the optimal threshold, as formalized below.

**Theorem 6.1 (Convergence of Stage 1, Informal)** *Assume rewards are bounded. Run projected gradient descent with a batch size of $N$. For both right- and left-CVaR cases, the optimality gap in stage 1 after $M$ steps is $\mathcal{O}\left(\frac{1}{\sqrt{M}} + \frac{1}{N}\right)$.*

The convergence bound reveals two distinct sources of error: an **optimization error** of order $\mathcal{O}(1/\sqrt{M})$, which reflects the standard convergence rate for convex optimization and vanishes as the iteration count $M \to \infty$; and a **statistical error** of order $\mathcal{O}(1/N)$, representing the irreducible error floor caused by the bias and variance of the ratio estimator, which vanishes as the batch size $N \to \infty$. Consequently, Stage 1 reliably provides a high-precision estimate of $t^*$, ensuring the stability of the subsequent fine-tuning stage. The complete statement and proof are deferred to Appendix B.4.

### 6.2 ANALYSIS OF STAGE 2

Since the optimal threshold $t^*$ is determined numerically in Stage 1, it inevitably contains a small approximation error. A critical theoretical question is whether this scalar error could lead to a catastrophic deviation in the target distribution constructed in Stage 2. We address this by establishing a bound on the Kullback-Leibler (KL) divergence between the ideal and approximate distributions.

**Theorem 6.2 (Sensitivity to Threshold Estimation)** *Let $\hat{t}$ be an approximation of the optimal threshold $t^*$ with estimation error $\delta = |\hat{t} - t^*|$. Let $p^*$ be the ideal target distribution defined by $t^*$, and $\hat{p}$ be the approximate distribution defined by $\hat{t}$. The KL divergence is bounded by $D_{KL}(p^*||\hat{p}) \leq \frac{2\delta}{\lambda_{\text{mode}}}$, where $\lambda_{\text{mode}} = \alpha(1 - \beta)$ if mode = Right, and $\lambda_{\text{mode}} = \alpha(1 - \beta)$ if mode = Left.*

Theorem 6.2 confirms the stability of our decoupled approach: a linearly small error in the threshold estimation leads to a linearly bounded error in the target distribution, scaled by $1/\alpha$. The proof can be found in Appendix B.5.

## 7  EXPERIMENTS

We evaluate the ability of TFFT in both risk-sensitive and novelty-seeking settings, and compare its performance against the pre-trained model (Pre-trained), Expected Fine-Tuning (EXP-FT) method, specifically Adjoint Matching (Domingo-Enrich et al., 2024), and FDC (De Santi et al., 2025), a recent state-of-the-art method for such problems. We present two types of experiments: (i) Illustrative, yet visually interpretable settings to provide insights about the different optimization dynamics, and (ii) High-dimensional real-world applications, namely (a) risk-averse image generation to control the worst-case generated samples, and (b) novelty-seeking molecular design to enhance the chances of discovering best-case single-point energy minimizers (Friede et al., 2024). Additional experimental details are provided in Appendix E.

### 7.1  ILLUSTRATIVE SETTINGS

To provide intuition for the behavior of CVaR-based generative optimization, we analyze a simple 2D example. We define the base distribution as a standard 2D normal distribution and the reward function as $r(x_1, x_2) = x_1 + x_2$. For this analysis, we directly compute the theoretical optimal distributions using importance sampling, assuming the generative model has sufficient capacity to represent them.

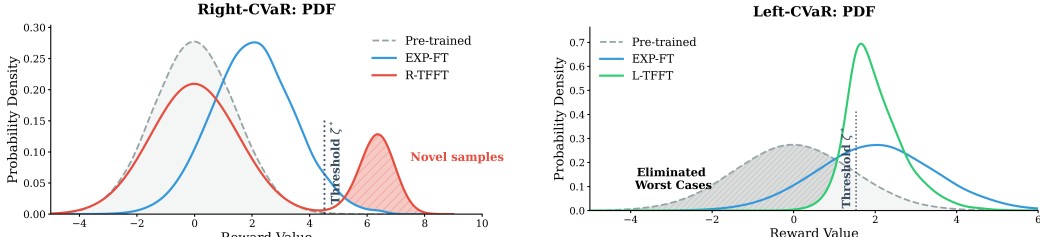

Figure 3: Probability Density Functions (PDFs) in the 2D example comparing Pre-trained, EXP-FT, and TFFT. (**Top**, right-CVaR, $\beta = 0.8$): R-TFFT concentrates probability mass in the high-reward tail, while EXP-FT shifts the mean. (**Bottom**, left-CVaR, $\beta = 0.2$): L-TFFT successfully truncates the lower tail, removing worst-case failures that persist in the EXP-FT distribution.

For the right-CVaR case, we set $\beta = 0.8$, targeting the largest 20% of the distribution. As shown in Fig. 3, EXP-FT uniformly shifts the mean of the distribution, whereas our right-CVaR fine-tuning method R-TFFT selectively amplifies the probability mass in the upper tail, demonstrating its ability to generate novel samples that exceed the performance threshold $\zeta^*$. For the left-CVaR case, we set $\beta = 0.2$, targeting the worst 20% of the distribution. As shown in Fig. 3, while EXP-FT improves the average reward, it retains the shape of the Gaussian tail, implying a non-negligible probability of suboptimal outcomes. In contrast, the left-CVaR fine-tuning method L-TFFT aggressively suppresses the probability of worst-case outcomes, thereby improving the model's worst-case performance.

**Efficiency Discussion.**  The efficiency gain of TFFT compared to FDC stems from avoiding the iterative distribution updates. As visualized in Fig. 2, FDC's distribution sequence $\{p^k\}$ oscillates significantly because the update rule depends on the current quantile $\mathrm{VaR}_\beta(p^k)$. This creates a *moving target* effect: a large distribution shift alters the threshold, which in turn changes the VaR value $\mathrm{VaR}_\beta(p^{k+1})$ and the effective pseudo-reward $[r(x) - \mathrm{VaR}_\beta(p^{k+1})]_+$. In contrast, TFFT bypasses this entirely by solving the dual form equation 5 offline using only the static prior $p^{\mathrm{pre}}$. By identifying the optimal threshold $\mathrm{VaR}_\beta(p^*)$ in advance, TFFT locks the target threshold, allowing the model to jump directly to the optimal solution in a single, stable fine-tuning step.

### 7.2  IMAGE GENERATION

To assess the efficacy of our L-TFFT in real-world applications, we apply it to high-dimensional text-to-image generation utilizing Stable-Diffusion v1-5 Rombach et al. (2022) as the backbone. We employ ImageReward (Xu et al., 2023) as the objective function, which is a general-purpose preference model trained on human feedback to quantify visual aesthetics and text-image alignment. For each fine-tuning method, we fix the computational budget to be 82 GPU hours and compare

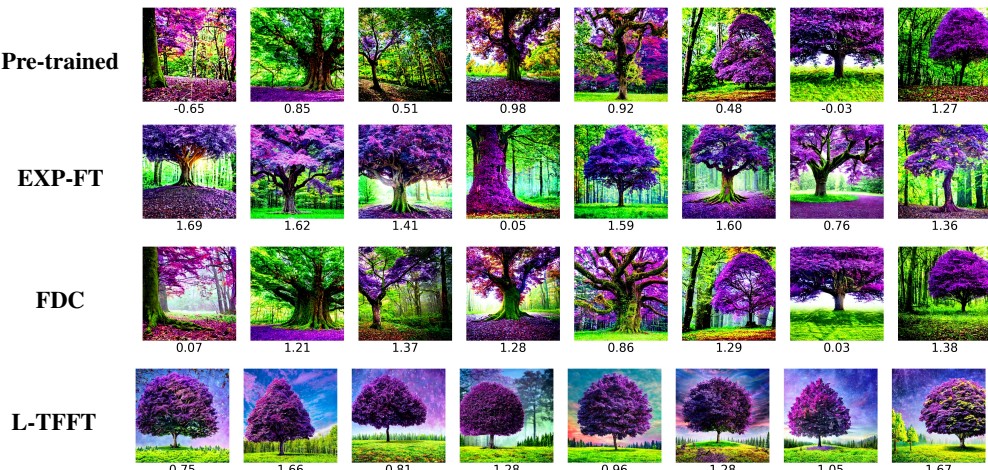

Figure 4: Qualitative comparison of generated samples for the prompt "A tree with purple leaves in a green forest". Each image is labeled with its ImageReward score. EXP-FT and FDC suffer from high variance and occasional failures. In contrast, L-TFFT produces consistently high-quality samples with a minimum score of 0.75, validating its ability to control the worst-case tail.

Table 2: Quantitative comparison of fine-tuning methods on Stable Diffusion maximizing ImageReward. Under a fixed computational budget, our method L-TFFT matches the state-of-the-art mean reward but significantly improves the worst-case metric L-CVaR$_{0.2}[r]$.

| | Training time GPU hours | $\mathbb{E}[r]$ ↑ | L-CVaR$_{0.2}[r]$ ↑ (bottom 20%) | CLIP-Score ↑ | HPS ↑ | DreamSim Var ↑ |
|---|---|---|---|---|---|---|
| Pre-trained | 0 | $0.271_{\pm 0.073}$ | $-1.147_{\pm 0.034}$ | $0.276_{\pm 0.003}$ | $0.255_{\pm 0.003}$ | $\mathbf{0.350}_{\pm 0.010}$ |
| EXP-FT | 82 | $\mathbf{0.852}_{\pm 0.065}$ | $-0.370_{\pm 0.035}$ | $0.279_{\pm 0.003}$ | $0.278_{\pm 0.026}$ | $0.306_{\pm 0.009}$ |
| FDC(K=2) | 82 | $0.633_{\pm 0.067}$ | $-0.604_{\pm 0.033}$ | $0.278_{\pm 0.003}$ | $0.269_{\pm 0.003}$ | $0.321_{\pm 0.009}$ |
| L-TFFT | 82 | $0.850_{\pm 0.063}$ | $\mathbf{-0.306}_{\pm 0.032}$ | $\mathbf{0.281}_{\pm 0.003}$ | $\mathbf{0.280}_{\pm 0.003}$ | $0.289_{\pm 0.008}$ |

their performance. Our evaluation metrics include ImageReward (and its CVaR value), ClipScore Hessel et al. (2021) for text-image consistency, HPSv2 Wu et al. (2023) for domain generalization, and DreamSim Fu et al. (2023) for diversity.

We set $\beta = 0.2$ to target the bottom 20% of the distribution. As summarized in Table 2, all three fine-tuning methods improve ImageReward values compared to the pre-trained baseline. However, under a fixed budget, FDC is less efficient and lags behind with a mean reward of 0.633. Crucially, while EXP-FT achieves the highest expected reward of 0.852, L-TFFT matches this expected performance (0.850) while significantly improving the worst-case metric L-CVaR$_{0.2}[r]$ from -0.370 (EXP-FT) to -0.306. This confirms L-TFFT's ability to raise the quality floor without sacrificing average performance. Their distribution curves are visualized in Fig. 10. The inverse CDF difference between L-TFFT and EXP-FT in the right panel highlights a distinct positive margin in the lower tail ($q < 0.2$), confirming that L-TFFT successfully raises the quality floor of worst-case samples compared to EXP-FT. When generating multiple samples for a fixed prompt, EXP-FT and FDC occasionally produces low-fidelity samples and suffers from high variance, while L-TFFT effectively precludes worst-case degradation, as shown in Fig. 4. Furthermore, our method demonstrates superior consistency and generalization, achieving the highest results on both CLIP-Score and HPSv2. While the slight decrease in DreamSim variance indicates a minor reduction in diversity, this reflects a necessary trade-off where the model suppresses diverse but poor-quality samples in the distribution tail.

## 7.3 MOLECULAR DESIGN

To demonstrate the efficacy of right-CVaR, we apply it to the task of molecular generation. We employ the continuous-time Markov chain (CTMC) implementation of FlowMol Dunn and Koes (2025) pretrained on the GEOM drugs dataset Axelrod and Gomez-Bombarelli (2022) as our base model. To optimize for chemical stability, we define the reward function as the negative xTB

Table 3: Quantitative comparison on molecular design. Our method R-TFFT achieves the highest right-CVaR values, indicating a superior ability to discover novel molecules in the distribution tail compared to FDC and EXP-FT. Crucially, it achieves this with $3\times$ less training time than FDC and maintains higher chemical validity than EXP-FT.

|  | Training time | FT calls | $\mathbb{E}[r]$ | R-CVaR$_{0.9}[r]$ | Validity | SA score |
|---|---|---|---|---|---|---|
| Pre-trained | 0 | 0 | $21.9_{\pm 1.21}$ | $161.5_{\pm 10.6}$ | $\textbf{87.4\%}_{\pm 0.12\%}$ | $7.64_{\pm 0.02}$ |
| EXP-FT | T | 1 | $\textbf{29.8}_{\pm 0.58}$ | $167.1_{\pm 7.22}$ | $78.3\%_{\pm 1.01\%}$ | $7.66_{\pm 0.02}$ |
| FDC (K=3) | $\approx 3T$ | 3 | $29.7_{\pm 0.85}$ | $172.5_{\pm 11.8}$ | $82.7\%_{\pm 0.47\%}$ | $\textbf{7.56}_{\pm 0.02}$ |
| R-TFFT | $\approx T$ | 1 | $27.1_{\pm 2.29}$ | $\textbf{183.4}_{\pm 15.8}$ | $85.7\%_{\pm 0.59\%}$ | $7.70_{\pm 0.04}$ |

energy, calculated using GFN1-xTB Friede et al. (2024). Our objective is to discover top-performing candidates rather than maximize the average sample reward. To this end, we employ TFFT with $\beta = 0.9$. We assess performance by sampling 2000 molecules from each model across three independent runs, verify their chemical validity using RDKit Sanitization (RDKit), and compute the energy distribution for the subset of molecules that pass this check.

As detailed in Table 3, R-TFFT achieves a superior right-CVaR of 183.4, significantly outperforming both EXP-FT (167.1) and the computationally heavier FDC (172.5). While FDC requires iterative updates totaling $\approx 3T$ training time, R-TFFT attains better results in a single fine-tuning step ($\approx T$), matching the computational cost of standard EXP-FT. Although EXP-FT yields the highest average reward, R-TFFT prioritizes discovery, generating valid, novel molecules in the high-reward tail (visualized in Fig. 5 ). In terms of chemical properties, R-TFFT maintains a high validity rate, outperforming both fine-tuning baselines and rivaling the pre-trained model. We attribute this to the modified reward function $[r - t]_+$, which effectively filters out low-reward samples. By limiting the number of samples contributing to model updates, the model stays close to the pre-trained model and preserves chemical validity. Regarding synthesizability, the Synthetic Accessibility (SA) score (Ertl and Schuffenhauer, 2009) still remains stable (7.70 vs 7.64 for pre-trained).

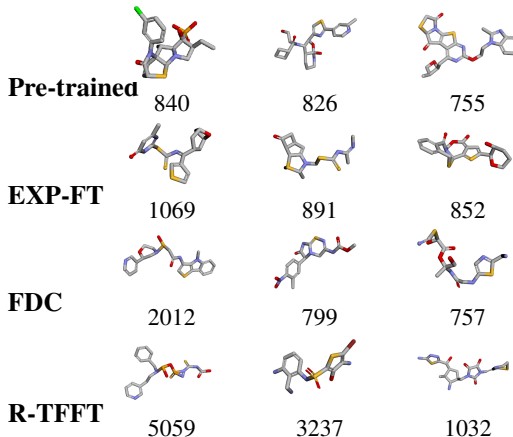

Figure 5: Visualization of the top-3 molecules with the highest reward found by each method. R-TFFT discovers candidates with significantly higher stability (scores up to 5059) compared to the best candidates from EXP-FT (1069) and FDC (2012), validating its ability to explore the extreme high-reward tail.

## 8 CONCLUSION

In this paper, we introduced TFFT, an efficient framework that enables to seek novel samples via Right-CVaR and mitigate worst cases via Left-CVaR. We decompose the CVaR generative optimization into a lightweight scalar search and a single fine-tuning step. This approach maintains a computational cost comparable to standard expected utility maximization. We validated TFFT on 2D, text-to-image, and molecular generation tasks, demonstrating that it efficiently and reliably shapes the reward tail. Future work could extend this dual decomposition approach to spectral risk measures for more kinds of distribution shaping.

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

## A    Related Works

**Fine-tuning of generative flow models.**   Fine-tuning pre-trained generative flow models with downstream reward functions has been rigorously formulated within the framework of stochastic optimal control Tang and Zhou (2024); Uehara et al. (2024a); Berner et al. (2022); Jensen et al. (2025); Wang et al. (2024); Blessing et al. (2025), where scalable solvers like Adjoint Matching Domingo-Enrich et al. (2024) have rendered high-dimensional control tractable. Similarly, recent work introduced fine-tuning schemes for constrained optimization of expected rewards Gutjahr et al. (2025). However, these methods are risk-neutral, focusing solely on maximizing the expected reward, rather than distributional measures. This formulation is often insufficient when utility is governed by the tail behavior of the reward distribution, whether for ensuring safety or facilitating the discovery of novel samples. An exception is FDC De Santi et al. (2025), which proposes a framework for optimizing general functionals of the generative density, including CVaR. However, FDC handles the non-linear CVaR objective by iteratively calling standard entropy-regularized solvers, which scales linearly with the number of iterations and is computationally expensive. In contrast, we demonstrate that for CVaR fine-tuning, this expensive iterative process can be bypassed entirely via a dual formulation, nearly matching the efficiency of standard expected reward maximization.

**Risk-sensitive learning.**   Risk-sensitive learning extends standard expected utility maximization to account for the variability and tail behavior of a distribution, and has been extensively explored in reinforcement learning and control Chow et al. (2018); Tamar et al. (2015); Urpí et al. (2021); Curi et al. (2020). Recently, these principles have been leveraged in generative modeling. For instance, Kim et al. (2025) employ diffusion models primarily as auxiliary samplers to synthesize informative data for the robust training of separate downstream tasks. Analogously, Foffano et al. (2025) leverage diffusion models as adversarial world models to generate worst-case trajectories for robust reinforcement learning. Unlike these approaches, which utilize generative models to synthesize training samples for downstream problems, our framework directly aims to control the risk of the generative model density itself. Similar schemes, although limited to the Left-CVaR formulation, have been proposed to mitigate the generation of toxic outputs in large language modeling (Chaudhary et al., 2024). In this work, we prioritize the use of Right-CVaR (i.e., novelty-seeking) to enhance discovery processes, while still supporting risk-aversion.

**Convex reinforcement learning.**   Convex RL (Hazan et al., 2019; Zahavy et al., 2021) generalizes RL to the maximization of concave functionals of the state distribution induced by a policy over a system's state space. This formulation, which includes the CVaR functional studied in this work (e.g., Mutti et al., 2022a; 2023), has been recently extended to settings where one wishes to maximize objectives beyond average rewards under a limited, finite budget of samples (e.g., De Santi et al., 2024b;a; Mutti et al., 2022b; Prajapat et al., 2023; Mutny et al., 2023). However, such methods are typically for non-combinatorial discrete spaces (Hazan et al., 2019; De Santi et al., 2024b) or for continuous representations (e.g., Liu and Abbeel, 2021) and cannot be straightforwardly applied to deep generative models.

## B    Proofs for Tail-aware Flow Fine-Tuning

### B.1    Proof of Theorem 4.1

Starting from the dual form of right-CVaR, we have:

$$
\max_{p^\pi} \ \text{R-CVaR}_{\beta, p^\pi}[r(X)] - \alpha D_{\text{KL}}(p^\pi \parallel p^{\text{pre}})
$$

$$
= \max_{p^\pi} \left\{ \min_t \left[ t + \frac{1}{1-\beta} \mathbb{E}_{p^\pi}[[r(X) - t]_+] \right] - \alpha D_{\text{KL}}(p^\pi \parallel p^{\text{pre}}) \right\}
$$

$$
= \min_t \max_{p^\pi} \left\{ t + \frac{1}{1-\beta} \mathbb{E}_{p^\pi}[[r(X) - t]_+] - \alpha D_{\text{KL}}(p^\pi \parallel p^{\text{pre}}) \right\},
$$

where the interchange of min and max follows from Sion's minimax theorem. For fixed $t$, the inner maximization over $p^\pi$ is a Linear GO problem with reward function $r_t(x) = \frac{[r(x)-t]_+}{1-\beta}$. The optimal

distribution is given by

$$q_{t,R}(x) \propto p^{\text{pre}}(x) \exp\left(\frac{r_t(x)}{\alpha}\right) = p^{\text{pre}}(x) \exp\left(\frac{[r(x) - t]_+}{\alpha(1 - \beta)}\right).$$

Substituting this optimal distribution back into the objective and using the fact that the optimal value of the inner problem is $\alpha \log \mathbb{E}_{p^{\text{pre}}}[\exp(r_t(X)/\alpha)]$, we obtain

$$\max_{p^\pi} \left\{ t + \frac{1}{1 - \beta} \mathbb{E}_{p^\pi}[[r(X) - t]_+] - \alpha D_{\text{KL}}(p^\pi \| p^{\text{pre}}) \right\}$$

$$= t + \alpha \log \mathbb{E}_{p^{\text{pre}}} \left[ \exp\left(\frac{[r(X) - t]_+}{\alpha(1 - \beta)}\right) \right].$$

Taking the minimum over $t$ completes the proof of equation 5, and equation 6 follows from the form of $q_{t,R}$ at the optimal $t^*$.

In what follows, we show that $t^*$ is exactly the VaR value of the target distribution $p_R^*$. For simplicity of notation, define

$$Z_R(t) \triangleq \mathbb{E}_{X \sim p^{\text{pre}}} \left[ \exp\left(\frac{[r(X) - t]_+}{\alpha(1 - \beta)}\right) \right] = \int_{\mathcal{X}} p^{\text{pre}}(x) \exp\left(\frac{[r(x) - t]_+}{\alpha(1 - \beta)}\right) dx,$$

and then $q_{t,R}$ can be written as

$$q_{t,R}(x) = \frac{1}{Z_R(t)} p^{\text{pre}}(x) \exp\left(\frac{[r(x) - t]_+}{\alpha(1 - \beta)}\right).$$

Since the reward distribution of the pre-trained model is continuous, we have $\mathbb{P}_{X \sim p^{\text{pre}}}(r(X) = t) = 0$. Then, it follows

$$Z_R'(t) = -\frac{1}{\alpha(1 - \beta)} \mathbb{E}_{X \sim p^{\text{pre}}} \left[ \exp\left(\frac{[r(X) - t]_+}{\alpha(1 - \beta)}\right) \mathbf{1}\{r(X) > t\} \right].$$

Define the objective in equation 5 as

$$F_R(t) = t + \alpha \log \mathbb{E}_{X \sim p^{\text{pre}}} \left[ \exp\left(\frac{[r(X) - t]_+}{\alpha(1 - \beta)}\right) \right] = t + \alpha \log Z_R(t).$$

Differentiating $F_R$ yields

$$F_R'(t) = 1 + \alpha \frac{d}{dt} \log Z_R(t) = 1 + \alpha \frac{Z_R'(t)}{Z_R(t)}$$

$$= 1 - \frac{1}{1 - \beta} \frac{\mathbb{E}_{X \sim p^{\text{pre}}} \left[ \exp\left(\frac{[r(X) - t]_+}{\alpha(1 - \beta)}\right) \mathbf{1}\{r(X) > t\} \right]}{\mathbb{E}_{X \sim p^{\text{pre}}} \left[ \exp\left(\frac{[r(X) - t]_+}{\alpha(1 - \beta)}\right) \right]}$$

$$= 1 - \frac{1}{1 - \beta} \mathbb{P}_{X \sim q_{t,R}}(r(X) > t),$$

where the lase equality follows from the definition of $q_{t,R}$. Since $t^*$ is the optimal solution of equation 5, we have $F_R'(t^*) = 0$. Then, we have

$$\mathbb{P}_{X \sim q_{t^\star,R}}(r(X) > t^\star) = \mathbb{P}_{X \sim p_R^\star}(r(X) > t^\star) = 1 - \beta,$$

which, by definition leads to $\text{VaR}_\beta(p_R^\star) = t^\star$. The proof is complete.

## B.2  PROOF OF THEOREM 4.2

This proof mirrors the logic in Appendix B.1. We present it here for completeness. Starting from the dual form of left-CVaR, we have

$$\max_{p^\pi} \text{L-CVaR}_{\beta, p^\pi}[r(X)] - \alpha D_{\text{KL}}(p^\pi \| p^{\text{pre}})$$

$$= \max_{p^\pi} \left\{ \max_t \left[ t - \frac{1}{\beta} \mathbb{E}_{p^\pi}[[t - r(X)]_+] \right] - \alpha D_{\text{KL}}(p^\pi \| p^{\text{pre}}) \right\}$$

$$= \max_t \max_{p^\pi} \left\{ t - \frac{1}{\beta} \mathbb{E}_{p^\pi}[[t - r(X)]_+] - \alpha D_{\text{KL}}(p^\pi \| p^{\text{pre}}) \right\}.$$

For every fixed $t$, the inner maximization over $p^\pi$ is a Linear GO problem, and the optimal distribution is given by

$$q_{t,L}(x) \propto p^{\text{pre}}(x) \exp\left(-\frac{[t - r(x)]_+}{\alpha\beta}\right).$$

Substituting this optimal distribution back into the objective and using the fact that the optimal value of the inner problem is $\alpha \log \mathbb{E}_{p^{\text{pre}}}[\exp(-[t - r(X)]_+/(\alpha\beta))]$, we obtain

$$\max_{p^\pi}\left\{t - \frac{1}{\beta}\mathbb{E}_{p^\pi}[[t - r(X)]_+] - \alpha D_{\text{KL}}(p^\pi \parallel p^{\text{pre}})\right\}$$
$$= t + \alpha \log \mathbb{E}_{p^{\text{pre}}}\left[\exp\left(-\frac{[t - r(X)]_+}{\alpha\beta}\right)\right].$$

Taking the maximum over $t$ completes the proof of equation 8, and equation 9 follows from the form of $q_{t,L}(x)$ at the optimal $t^*$.

In what follows, we show that $t^*$ is exactly the VaR value of the target distribution $p_L^*$. Define

$$Z_L(t) = \mathbb{E}_{X\sim p^{\text{pre}}}\left[\exp\left(-\frac{[t - r(X)]_+}{\alpha\beta}\right)\right],$$

and then $q_{t,L}$ can be written as

$$q_{t,L}(x) = \frac{1}{Z_L(t)}p^{\text{pre}}(x) \exp\left(-\frac{[t - r(x)]_+}{\alpha\beta}\right).$$

Differentiating $Z_L(t)$ yields

$$Z_L'(t) = \mathbb{E}_{p^{\text{pre}}}\left[-\frac{1}{\alpha\beta}\mathbf{1}(r(X) < t)\exp\left(-\frac{[t - r(X)]_+}{\alpha\beta}\right)\right].$$

Define the objective in equation 8 as

$$F_L(t) = t + \alpha \log Z_L(t).$$

Differentiating $F_L$ yields

$$F_L'(t) = 1 + \alpha\frac{Z_L'(t)}{Z_L(t)}$$
$$= 1 + \alpha\frac{\mathbb{E}_{p^{\text{pre}}}\left[-\frac{1}{\alpha\beta}\mathbf{1}(r(X) < t)\exp\left(-\frac{[t-r(X)]_+}{\alpha\beta}\right)\right]}{Z_L(t)}$$
$$= 1 - \frac{1}{\beta}\int \mathbf{1}(r(x) < t)\frac{p^{\text{pre}}(x)\exp\left(-\frac{[t-r(x)]_+}{\alpha\beta}\right)}{Z_L(t)}dx$$
$$= 1 - \frac{1}{\beta}\mathbb{P}_{X\sim q_{t,L}}(r(X) < t),$$

where the lase equality follows from the definition of $q_{t,L}$. Since $t^*$ is the optimal solution of equation 8, we have $F_L'(t^*) = 0$, which yields

$$\mathbb{P}_{q_{t,L}^*}(r(X) < t^*) = \mathbb{P}_{p_L^*}(r(X) < t^*) = \beta.$$

According to the definition of VaR, we have $\text{VaR}_\beta(p_L^*) = t^*$. The proof is complete.

### B.3   PROOF OF THEOREM 5.1

We first derive the properties for the right-CVaR minimization case. Recall that

$$F_R(t) = t + \alpha \log \mathbb{E}_{x\sim p_1^{pre}}\left[\exp\left(\frac{[r(x) - t]_+}{\alpha(1 - \beta)}\right)\right].$$

Let $Z_R(t) = \mathbb{E}_{x \sim p_1^{pre}} \left[ \exp \left( \frac{[r(x) - t]_+}{\alpha(1-\beta)} \right) \right]$. The derivative with respect to $t$ is:

$$F_R'(t) = 1 + \frac{\alpha}{Z_R(t)} \frac{d}{dt} \mathbb{E} \left[ \exp \left( \frac{[r(x) - t]_+}{\alpha(1-\beta)} \right) \right].$$

Using the property $\frac{d}{dt}[r(x) - t]_+ = -\mathbf{1}(r(x) > t)$, where $\mathbf{1}$ is the indicator function, we obtain

$$F_R'(t) = 1 - \frac{1}{1-\beta} \frac{\mathbb{E}_{x \sim p_1^{pre}} \left[ \mathbf{1}(r(x) > t) \exp \left( \frac{[r(x) - t]_+}{\alpha(1-\beta)} \right) \right]}{Z_R(t)}.$$

By defining the tilted distribution $q_{t,R}(x) \propto p_1^{pre}(x) \exp \left( \frac{[r(x) - t]_+}{\alpha(1-\beta)} \right)$, the gradient can be expressed as:

$$F_R'(t) = 1 - \frac{1}{1-\beta} \mathbb{E}_{x \sim q_{t,R}}[\mathbf{1}(r(x) > t)].$$

To establish convexity and smoothness, we differentiate $F_R'(t)$ with respect to $t$, which yields $F_R''(t) = -\frac{1}{1-\beta} \int \mathbf{1}(r(x) > t) \frac{d}{dt} q_{t,R}(x) dx$. Note that

$$\frac{d}{dt} q_{t,R}(x) = \frac{1}{\alpha(1-\beta)} q_{t,R}(x) \left( \mathbb{E}_{q_{t,R}}[\mathbf{1}(r(x) > t)] - \mathbf{1}(r(x) > t) \right).$$

Then, we obtain

$$\begin{aligned}
F_R''(t) &= -\frac{1}{1-\beta} \int \mathbf{1}(r(x) > t) \left[ \frac{1}{\alpha(1-\beta)} q_{t,R}(x) \left( \mathbb{E}_{q_{t,R}}[\mathbf{1}(r(x) > t)] - \mathbf{1}(r(x) > t) \right) \right] dx \\
&= \frac{1}{\alpha(1-\beta)^2} \left( \mathbb{E}_{q_{t,R}}[\mathbf{1}(r(x) > t)^2] - \left( \mathbb{E}_{q_{t,R}}[\mathbf{1}(r(x) > t)] \right)^2 \right) \\
&= \frac{1}{\alpha(1-\beta)^2} \text{Var}_{x \sim q_{t,R}}[\mathbf{1}(r(x) > t)]
\end{aligned}$$

(12)

Since variance is non-negative and $\alpha(1-\beta)^2 > 0$, $F_R''(t) \geq 0$ for all $t$, proving that $F_R(t)$ is strictly convex.

Since the variance of any indicator variable $\mathbf{1} \in \{0, 1\}$ is strictly bounded by $\frac{1}{4}$, we have

$$|F_R''(t)| \leq \frac{1}{\alpha(1-\beta)^2} \cdot \frac{1}{4} = L_R.$$

This establishes that $F_R(t)$ is smooth with the parameter $L_R$.

For left-CVaR optimization, the objective function to maximize is:

$$F_L(t) = t + \alpha \log \mathbb{E}_{x \sim p^{pre}} \left[ \exp \left( -\frac{[t - r(x)]_+}{\alpha\beta} \right) \right].$$

Similarly, we have

$$F_L'(t) = 1 - \frac{1}{\beta} \frac{\mathbb{E} \left[ \mathbf{1}(r(x) < t) \exp \left( -\frac{[t - r(x)]_+}{\alpha\beta} \right) \right]}{\mathbb{E} \left[ \exp \left( -\frac{[t - r(x)]_+}{\alpha\beta} \right) \right]}.$$

By defining the tilted distribution $q_{t,L}(x) \propto p^{pre}(x) \exp \left( -\frac{[t - r(x)]_+}{\alpha\beta} \right)$, we get:

$$F_L'(t) = 1 - \frac{1}{\beta} \mathbb{E}_{x \sim q_{t,L}}[\mathbf{1}(r(x) < t)].$$

Differentiating again with respect to $t$ using the same density shift logic as the right-CVaR case, we obtain

$$\begin{aligned}
F_L''(t) &= -\frac{1}{\beta} \frac{d}{dt} \mathbb{E}_{x \sim q_{t,L}}[\mathbf{1}(r(x) < t)] \\
&= -\frac{1}{\alpha\beta^2} \text{Var}_{x \sim q_{t,L}}[\mathbf{1}(r(x) < t)].
\end{aligned}$$

Since the variance is always non-negative, we have $F_L''(t) \leq 0$, proving that $F_L(t)$ is strictly concave. Applying the upper bound for the variance of an indicator variable, we get

$$|F_L''(t)| \leq \frac{1}{\alpha\beta^2} \cdot \frac{1}{4} := L_L.$$

This establishes that $F_L(t)$ is smooth with the parameter $L_L$. The proof is complete.

## B.4 THEOREM 6.1 AND ITS PROOF

In this section, we provide the formal theorem for the convergence of stage 1 as well as its proof. Recalling the definitions of $F_R$ and $F_L$ in equation 10 and equation 11, their gradients are given by

$$F'_R(t) = 1 - \frac{1}{1-\beta} \frac{\mathbb{E}_{x \sim p^{\text{pre}}}\left[\mathbf{1}(r(x) > t) \exp\left(\frac{[r(x)-t]_+}{\alpha(1-\beta)}\right)\right]}{\mathbb{E}_{x \sim p^{\text{pre}}}\left[\exp\left(\frac{[r(x)-t]_+}{\alpha(1-\beta)}\right)\right]},$$

$$F'_L(t) = 1 - \frac{1}{\beta} \frac{\mathbb{E}_{x \sim p^{\text{pre}}}\left[\mathbf{1}(r(x) < t) \exp\left(-\frac{[t-r(x)]_+}{\alpha\beta}\right)\right]}{\mathbb{E}_{x \sim p^{\text{pre}}}\left[\exp\left(-\frac{[t-r(x)]_+}{\alpha\beta}\right)\right]}.$$

In what follows, we present the formal version of Theorem 6.1.

**Theorem 6.1** *Suppose that we are given $N$ samples $x_i$, $i = 1, \ldots\ldots, N$. We construct the gradient estimators*

$$\hat{g}_N^R(t) = 1 - \frac{1}{1-\beta} \frac{\sum_{i=1}^N \mathbf{1}(r(x_i) > t) \exp\left(\frac{[r(x_i)-t]_+}{\alpha(1-\beta)}\right)}{\sum_{i=1}^N \exp\left(\frac{[r(x_i)-t]_+}{\alpha(1-\beta)}\right)}, \tag{13}$$

$$\hat{g}_N^L(t) = 1 - \frac{1}{\beta} \frac{\sum_{i=1}^N \mathbf{1}(r(x_i) < t) \exp\left(-\frac{[t-r(x_i)]_+}{\alpha\beta}\right)}{\sum_{i=1}^N \exp\left(-\frac{[t-r(x_i)]_+}{\alpha\beta}\right)}. \tag{14}$$

*for $F'_R(t)$ and $F'_L(t)$, respectively. Assume rewards lie in a bounded interval $\mathcal{I}$. For right-CVaR, let the update rule $t_{k+1} = \Pi_{\mathcal{I}}(t_k - \eta \hat{g}_N^R(t_k))$, then we have $\mathbb{E}[F_R(\bar{t}_M) - F_R(t^*)] \leq \mathcal{O}\left(\frac{1}{\sqrt{M}}\right) + \mathcal{O}\left(\frac{1}{N}\right)$. For the left-CVaR, let the update rule $t_{k+1} = \Pi_{\mathcal{I}}(t_k + \eta \hat{g}_N^L(t_k))$, then we have $\mathbb{E}[F_L(t^*) - F_L(\bar{t}_M)] \leq \mathcal{O}\left(\frac{1}{\sqrt{M}}\right) + \mathcal{O}\left(\frac{1}{N}\right)$.*

To prove Theorem 6.1, we establish general lemmas regarding ratio estimators and biased SGD, and then apply them to the specific geometry of right- and left-CVaR.

### B.4.1 PROPERTIES OF THE RATIO GRADIENT ESTIMATOR

Both CVaR gradients take the form of a ratio of expectations. We first bound the bias and variance of the sample-based approximation.

**Lemma B.1 (Bias and Variance of Ratio Estimators.)** *Let $U, V$ be random variables such that $0 \leq U \leq V$ almost surely and $V \in [m, M]$ almost surely for constants $0 < m \leq M$. Let $\{(U_i, V_i)\}_{i=1}^N$ be i.i.d. copies. Define $\bar{U} = \frac{1}{N}\sum U_i$ and $\bar{V} = \frac{1}{N}\sum V_i$. Let $\rho = \mathbb{E}[U]/\mathbb{E}[V]$ and $\hat{\rho} = \bar{U}/\bar{V}$. Let $\Xi := M/m \geq 1$. Then:*

$$|\mathbb{E}[\hat{\rho}] - \rho| \leq \frac{\Xi^3}{4N}, \quad \text{and} \quad Var(\hat{\rho}) \leq \frac{\Xi^4}{N}.$$

**Proof:** First, we normalize the variables to simplify the bounds. Define $U' = U/m$ and $V' = V/m$. It follows that $0 \leq U' \leq V'$ and $V' \in [1, \Xi]$ almost surely. Note that the ratio remains invariant: $\hat{\rho} = \bar{U}/\bar{V} = \bar{U}'/\bar{V}'$ and $\rho = \mathbb{E}[U']/\mathbb{E}[V']$. Thus, it suffices to prove the bounds for the rescaled variables. Let $\mu_U = \mathbb{E}[U']$ and $\mu_V = \mathbb{E}[V']$. Note that $\mu_V \geq 1$ and $\mu_U \leq \Xi$.

We decompose the error by adding and subtracting terms:

$$\mathbb{E}\left[\frac{\bar{U}'}{\bar{V}'}\right] - \frac{\mu_U}{\mu_V} = \mathbb{E}\left[\frac{\bar{U}'}{\bar{V}'} - \frac{\bar{U}'}{\mu_V} + \frac{\bar{U}'}{\mu_V} - \frac{\mu_U}{\mu_V}\right] = \mathbb{E}\left[\bar{U}'\left(\frac{1}{\bar{V}'} - \frac{1}{\mu_V}\right)\right].$$

We can further rewrite the expectation of the product as the covariance plus the product of expectations

$$\mathbb{E}\left[\bar{U}'\left(\frac{1}{\bar{V}'} - \frac{1}{\mu_V}\right)\right] = \mathbb{E}\left[(\bar{U}' - \mu_U)\left(\frac{1}{\bar{V}'} - \frac{1}{\mu_V}\right)\right] + \mu_U\left(\mathbb{E}\left[\frac{1}{\bar{V}'}\right] - \frac{1}{\mu_V}\right).$$

For the first term, by the Cauchy-Schwarz inequality, we have

$$\left| \mathbb{E}\left[ (\bar{U}' - \mu_U)\left( \frac{1}{\bar{V}'} - \frac{1}{\mu_V} \right) \right] \right| \leq \sqrt{\text{Var}(\bar{U}')\text{Var}\left( \frac{1}{\bar{V}'} \right)}.$$

Since the function $\phi(v) = 1/v$ has derivative $\phi'(v) = -1/v^2$ and $|\phi'(v)| \leq 1$ for $v \geq 1$, $\phi$ is 1-Lipschitz on $[1, \infty)$. For a $L$−Lipschitz continuous function $f$, we have $\text{Var}(f(X)) \leq L^2\text{Var}(X)$. Therefore, $\text{Var}(1/\bar{V}') \leq \text{Var}(\bar{V}')$. Also, for bounded random variables in $[1, \Xi]$, the variance of the sample mean is bounded by $\text{Var}(\bar{V}') \leq \frac{(\Xi-1)^2}{4N}$. Similarly, since $U' \leq V' \leq \Xi$, $\text{Var}(\bar{U}') \leq \frac{\Xi^2}{4N}$. Thus, we have

$$\sqrt{\text{Var}(\bar{U}')\text{Var}\left( \frac{1}{\bar{V}'} \right)} \leq \sqrt{\frac{\Xi^2}{4N} \cdot \frac{(\Xi-1)^2}{4N}} = \frac{\Xi(\Xi-1)}{4N}.$$

For the second term, we have

$$\mu_U\left( \mathbb{E}\left[ \frac{1}{\bar{V}'} \right] - \frac{1}{\mu_V} \right) \leq \mu_U\text{Var}\left( \frac{1}{\bar{V}'} \right) \leq \mu_U\text{Var}(\bar{V}') \leq \mu_U\frac{(\Xi-1)^2}{4N} \leq \Xi\frac{(\Xi-1)^2}{4N}.$$

Summing the two terms, we obtain

$$|\mathbb{E}[\hat{\rho}] - \rho| \leq \frac{\Xi(\Xi-1)}{4N} + \frac{\Xi(\Xi-1)^2}{4N} = \frac{\Xi(\Xi-1)}{4N}(1 + \Xi - 1) = \frac{\Xi^2(\Xi-1)}{4N} \leq \frac{\Xi^3}{4N}.$$

In what follows, we bound the variance. For $u, u' \in [0, \Xi]$ and $v, v' \in [1, \Xi]$, we have

$$\left| \frac{u}{v} - \frac{u'}{v'} \right| = \left| \frac{uv' - u'v}{vv'} \right| = \left| \frac{v'(u - u') + u'(v' - v)}{vv'} \right| \leq \frac{|u - u'|}{|v|} + \frac{|u'(v' - v)|}{|vv'|} \leq |u-u'|+\Xi|v-v'|.$$

We replace the generic variables $u, v$ with our random sample means $\bar{U}', \bar{V}'$ and the constants $u', v'$ with the true means $\mu_U, \mu_V$, which yields

$$\left| \frac{\bar{U}'}{\bar{V}'} - \frac{\mu_U}{\mu_V} \right| \leq |\bar{U}' - \mu_U| + \Xi|\bar{V}' - \mu_V|.$$

Then, we have

$$\text{Var}\left( \frac{\bar{U}'}{\bar{V}'} \right) \leq \mathbb{E}\left[ \left( \frac{\bar{U}'}{\bar{V}'} - \frac{\mu_U}{\mu_V} \right)^2 \right] \leq 2\mathbb{E}[(\bar{U}'-\mu_U)^2]+2\Xi^2\mathbb{E}[(\bar{V}'-\mu_V)^2] \leq 2\text{Var}(\bar{U}')+2\Xi^2\text{Var}(\bar{V}')$$

Substituting the variance bounds $\text{Var}(\bar{U}') \leq \frac{\Xi^2}{4N}$ and $\text{Var}(\bar{V}') \leq \frac{(\Xi-1)^2}{4N}$, we have

$$\text{Var}(\hat{\rho}) \leq 2\left( \frac{\Xi^2}{4N} \right) + 2\Xi^2\left( \frac{\Xi^2}{4N} \right) \leq \frac{\Xi^4}{N}.$$

The proof is complete. $\qquad\qquad\square$

### B.4.2 CONVERGENCE OF BIASED PROJECTED SGD

We derive a convergence bound for minimizing a smooth convex function using a biased gradient oracle.

**Lemma B.2** *Let $f : \mathbb{R} \to \mathbb{R}$ be convex and $L$-smooth with minimizer $t^*$. Fix an interval $\mathcal{I}$ containing $t^*$ with diameter $D$. Let the update be $t_{k+1} = \Pi_{\mathcal{I}}(t_k - \eta\hat{g}_k)$, where $\hat{g}_k = f'(t_k) + e_k$. Assume $|\mathbb{E}[e_k|t_k]| \leq \epsilon$ and $\mathbb{E}[e_k^2|t_k] \leq \sigma^2 + \epsilon^2$. If $\eta \leq \frac{1}{4L}$, then we have*

$$\mathbb{E}[f(\bar{t}_M) - f(t^*)] \leq \frac{(t_0 - t^*)^2}{\eta M} + 2\eta(\sigma^2 + \epsilon^2) + 2D\epsilon.$$

**Proof:** By the non-expansiveness of the Euclidean projection $\Pi_{\mathcal{I}}$, we have:

$$|t_{k+1} - t^*|^2 \le |t_k - \eta \hat{g}_k - t^*|^2 = |t_k - t^*|^2 - 2\eta(t_k - t^*)\hat{g}_k + \eta^2 \hat{g}_k^2.$$

Substituting $\hat{g}_k = f'(t_k) + e_k$ yields

$$|t_{k+1} - t^*|^2 \le |t_k - t^*|^2 - 2\eta(t_k - t^*)f'(t_k) - 2\eta(t_k - t^*)e_k + \eta^2(f'(t_k) + e_k)^2.$$

By convexity, $f(t_k) - f(t^*) \le f'(t_k)(t_k - t^*)$. Rearranging the inequality to isolate the function value gap, we obtain

$$
\begin{aligned}
2\eta(f(t_k) - f(t^*)) \le & |t_k - t^*|^2 - |t_{k+1} - t^*|^2 - 2\eta(t_k - t^*)e_k + \eta^2(f'(t_k) + e_k)^2 \\
\le & |t_k - t^*|^2 - |t_{k+1} - t^*|^2 - 2\eta(t_k - t^*)e_k + 2\eta^2(f'(t_k)^2 + 2\eta^2(e_k)^2 \\
\le & |t_k - t^*|^2 - |t_{k+1} - t^*|^2 - 2\eta(t_k - t^*)e_k + 4\eta^2(L(f(t_k) - f(t^*)) + 2\eta^2(e_k)^2,
\end{aligned}
$$

where the last inequality follows since $(f'(t))^2 \le 2L(f(t) - f(t^*))$. Rearranging and taking expectations on both sides, we have

$$\mathbb{E}[f(t_k) - f(t^*)] \le \frac{\mathbb{E}[|t_k - t^*|^2] - \mathbb{E}[|t_{k+1} - t^*|^2]}{2\eta} + D\epsilon + \frac{\eta}{2}\left(4L\mathbb{E}[f(t_k) - f(t^*)] + 2(\sigma^2 + \epsilon^2)\right).$$

Rearranging terms yields

$$(1 - 2\eta L)\mathbb{E}[f(t_k) - f(t^*)] \le \frac{\mathbb{E}[|t_k - t^*|^2] - \mathbb{E}[|t_{k+1} - t^*|^2]}{2\eta} + \eta(\sigma^2 + \epsilon^2) + D\epsilon$$

Assuming $\eta \le \frac{1}{4L}$, we have $1 - 2\eta L \ge 1/2$. Therefore, we have

$$\mathbb{E}[f(t_k) - f(t^*)] \le \frac{\mathbb{E}[|t_k - t^*|^2] - \mathbb{E}[|t_{k+1} - t^*|^2]}{\eta} + 2\eta(\sigma^2 + \epsilon^2) + 2D\epsilon.$$

Summing up yields

$$\frac{1}{M} \sum_{k=0}^{M-1} \mathbb{E}[f(t_k) - f(t^*)] \le \frac{|t_0 - t^*|^2}{\eta M} + 2\eta(\sigma^2 + \epsilon^2) + 2D\epsilon.$$

By Jensen's inequality, $f(\bar{t}_M) - f(t^*) \le \frac{1}{M}\sum(f(t_k) - f(t^*))$, which completes the proof. $\qquad\square$

### B.4.3 PROOF OF THEOREM 6.1

We now apply the above results to our specific Stage 1 objectives. Assume the reward is bounded $r(x) \in [r_{min}, r_{max}]$.

**Right-CVaR (Minimization).** The objective $F_R(t)$ is convex and $L_R$-smooth (Theorem 5.1). The gradient is $F_R'(t) = 1 - \frac{1}{1-\beta}\frac{A(t)}{B(t)}$, where $A(t) = \mathbb{E}[\exp([r(X) - t]_+ \mathbf{1}(r(x) > t)/(\alpha(1 - \beta)))]$, $B(t) = \mathbb{E}[\exp([r(X) - t]_+/(\alpha(1 - \beta)))]$. Since rewards are bounded, $B(t)$ is bounded away from zero for $t$ in a compact range (specifically $B(t) \ge 1$ as the exponent is non-negative). The numerator $A(t)$ is similarly bounded. Thus, by Lemma B.1, the gradient estimator $\hat{g}_N^R$ has bias $\epsilon = \mathcal{O}(1/N)$ and variance $\sigma^2 = \mathcal{O}(1/N)$. Applying Lemma B.2 with $\eta = \frac{1}{\sqrt{M}}$, we have

$$\mathbb{E}[F_R(\bar{t}_M) - F_R(t^*)] \le \mathcal{O}\left(\frac{1}{\sqrt{M}}\right) + \mathcal{O}\left(\frac{1}{N}\right).$$

**left-CVaR (Maximization).** The objective $F_L(t)$ is concave. The gradient term involves $B_L(t) = \mathbb{E}[\exp(-[t - r(X)]_+/\alpha\beta)]$. Since rewards are bounded, the exponent is bounded below, ensuring $B_L(t) \ge e^{-(r_{max}-r_{min})/(\alpha\beta)} > 0$. The conditions of Lemma B.1 are satisfied. Applying Lemma B.2 to $-F_L$ yields the analogous convergence result.

### B.5 PROOF OF THEOREM 6.2

We prove the sensitivity bound for the right-CVaR case in detail. The result for left-CVaR follows from symmetric arguments.

Let $t^*$ be the optimal threshold and $\hat{t}$ be the numerical approximation such that $\delta = |t^* - \hat{t}|$. Recall that for right-CVaR, the optimal distribution $p_R^*$ and the approximate distribution $\hat{p}_R$ are defined as:

$$p_R^*(x) = \frac{1}{Z_R(t^*)} p^{\text{pre}}(x) \exp\left(\frac{[r(x) - t^*]_+}{\alpha(1-\beta)}\right),$$

$$\hat{p}_R(x) = \frac{1}{Z_R(\hat{t})} p^{\text{pre}}(x) \exp\left(\frac{[r(x) - \hat{t}]_+}{\alpha(1-\beta)}\right),$$

where $Z_R(t) = \mathbb{E}_{x \sim p^{\text{pre}}}\left[\exp\left(\frac{[r(x) - t]_+}{\alpha(1-\beta)}\right)\right]$. The Kullback-Leibler (KL) divergence is given by:

$$D_{\text{KL}}(p_R^* \| \hat{p}_R) = \mathbb{E}_{x \sim p^*}\left[\ln p_R^*(x) - \ln \hat{p}_R(x)\right]$$

$$= \mathbb{E}_{x \sim p_R^*}\left[\frac{[r(x) - t^*]_+}{\alpha(1-\beta)} - \ln Z_R(t^*) - \left(\frac{[r(x) - \hat{t}]_+}{\alpha(1-\beta)} - \ln Z_R(\hat{t})\right)\right]$$

$$= \underbrace{\frac{1}{\alpha(1-\beta)} \mathbb{E}_{x \sim p_R^*}\left[[r(x) - t^*]_+ - [r(x) - \hat{t}]_+\right]}_{(A)} + \underbrace{\ln Z_R(\hat{t}) - \ln Z_R(t^*)}_{(B)}. \quad (15)$$

**Term (A):** Since $|[a]_+ - [b]_+| \le |a - b|$, we have

$$(A) \le \frac{1}{\alpha(1-\beta)} \mathbb{E}_{x \sim p_R^*}[\delta] = \frac{\delta}{\alpha(1-\beta)}.$$

**Term (B):** Using the Lipschitz property, for any $x$:

$$\left|[r(x) - t^*]_+ - [r(x) - \hat{t}]_+\right| \le \left|(r(x) - t^*) - (r(x) - \hat{t})\right| = |\hat{t} - t^*| = \delta.$$

We also have

$$[r(x) - t^*]_+ - \delta \le [r(x) - \hat{t}]_+ \le [r(x) - t^*]_+ + \delta.$$

We substitute these inequalities into the definition of the partition function $Z(\hat{t})$:

$$Z_R(\hat{t}) = \mathbb{E}_{x \sim p^{\text{pre}}}\left[\exp\left(\frac{[r(x) - \hat{t}]_+}{\alpha(1-\beta)}\right)\right]$$

$$\le \mathbb{E}_{x \sim p^{\text{pre}}}\left[\exp\left(\frac{[r(x) - t^*]_+ + \delta}{\alpha(1-\beta)}\right)\right]$$

$$= e^{\delta/\alpha(1-\beta)} Z_R(t^*).$$

Taking the natural logarithm yields $\ln Z_R(\hat{t}) - \ln Z_R(t^*) \le \frac{\delta}{\alpha(1-\beta)}$. Similarly, using the lower bound $[r(x) - \hat{t}]_+ \ge [r(x) - t^*]_+ - \delta$, we obtain $\ln Z_R(\hat{t}) - \ln Z_R(t^*) \ge -\frac{\delta}{\alpha(1-\beta)}$. Thus, we have

$$|(B)| \le \frac{\delta}{\alpha(1-\beta)}.$$

Substituting the bounds for (A) and (B) back into equation 15, we get

$$D_{\text{KL}}(p_R^* \| \hat{p}) \le \frac{2\delta}{\alpha(1-\beta)}.$$

For the left-CVaR case, it is symmetric to the right-CVaR case and omitted here. The proof is complete.

## C   DETAILED ALGORITHM

To ensure completeness, below we provide a detailed pseudocode for our TFFT based on the adjoint matching Domingo-Enrich et al. (2024) framework.

---

**Algorithm 2** Detailed Tail-aware Flow Fine-Tuning

---

1: **Input:** Pre-trained FM velocity field $u^{\text{pre}}$, step size $h$, number of fine-tuning iterations $N$, reward $r$, coefficient $\alpha$ as in equation 3, and the mode $M \in \{\text{Right}, \text{Left}\}$.
2: *// Stage 1: Optimal Threshold Optimization*
3: **if** $M = \text{Right}$ **then**
4:   Set Search Goal: $\min_t F_R(t)$.
5: **else**
6:   Set Search Goal: $\max_t F_L(t)$.
7: **end if**
8: Use numerical methods like gradient ascent
9: **Return** $t^*$
10: *// Stage 2: Single Fine-Tuning Step*
11: **if** $M = \text{Right}$ **then**
12:   Construct pseudo-reward: $r^*(x) = \frac{[r(x)-t^*]_+}{1-\beta}$.
13: **else**
14:   Construct pseudo-reward: $r^*(x) = -\frac{[t^*-r(x)]_+}{\beta}$.
15: **end if**
16: Initialize fine-tuned vector fields: $u^{\text{finetune}}_\theta = u^{\text{pre}}$ with parameters $\theta$.
17: **for** $n \in \{0, \ldots, N-1\}$ **do**
18:   Sample $m$ trajectories $\boldsymbol{X} = (X_t)_{t \in \{0,\ldots,1\}}$ with memoryless noise schedule $\sigma(t)$ Domingo-Enrich et al. (2024), *e.g.*:

$$X_{t+h} = X_t + h\left(2u^{\text{finetune}}_\theta(X_t, t) - \frac{\dot{\alpha}_t}{\alpha_t}X_t\right) + \sqrt{h}\sigma(t)\varepsilon_t, \quad \varepsilon_t \sim \mathcal{N}(0, I), \quad X_0 \sim \mathcal{N}(0, I). \tag{16}$$

19:   For each trajectory, solve the *lean adjoint ODE* backwards in time from $t = 1$ to $0$, *e.g.*:

$$\tilde{a}_{t-h} = \tilde{a}_t + h\tilde{a}_t^\top \nabla_{X_t}\left(2u^{\text{pre}}(X_t, t) - \frac{\dot{\alpha}_t}{\alpha_t}X_t\right), \quad \tilde{a}_1 = -\nabla_{X_1}r^*(X_1). \tag{17}$$

20:   Note that $X_t$ and $\tilde{a}_t$ should be computed without gradients, *i.e.*, $X_t = \texttt{stopgrad}(X_t)$, $\tilde{a}_t = \texttt{stopgrad}(\tilde{a}_t)$.
21:   For each trajectory, compute the Adjoint Matching objective:

$$\mathcal{L}(\theta) = \sum_{t \in \{0,\ldots,1-h\}} \left\| \frac{2}{\sigma(t)}(u^{\text{finetune}}_\theta(X_t, t) - u^{\text{pre}}(X_t, t)) + \sigma(t)\tilde{a}_t \right\|^2. \tag{18}$$

22:   Compute the gradient $\nabla_\theta \mathcal{L}(\theta)$ and update $\theta$ using favorite gradient descent algorithm.
23: **end for**
24: **Output:** Fine-tuned vector field $u^{\text{finetune}}_\theta$

---

# D  ADDITIONAL RESULTS ON FDC

## D.1  COMPARISON TO NON-LINEAR GENERATIVE OPTIMIZATION

In this section, we clarify the relationship between TFFT and FDC within the context of tail-aware GO. Below we use right-CVaR as an example.

Specifically, at iteration $k$, FDC updates the distribution by solving the following linear GO problem

$$p^{\pi_{k+1}} = \arg\max_{p^\pi} \langle \delta\mathcal{G}(p^{\pi_k}), p^\pi \rangle - \eta D_{\mathrm{KL}}(p^\pi || p^{\pi_k}), \tag{19}$$

where $\mathcal{G}$ is defined in equation 3, $\eta > 0$ is a step-size parameter, and $p^{\pi_0} = p^{\mathrm{pre}}$. Here, $\delta\mathcal{G}$ denotes the first variation, which can be interpreted as an infinite-dimensional gradient in the space of probability distributions. Assuming that equation 19 is solved exactly, the updated distributions of FDC satisfy

$$p^{\pi_{k+1}}(x) \propto \left(p^{\pi_k}(x)\right)^{1-\alpha/\eta} \left(p^{\mathrm{pre}}(x)\right)^{\alpha/\eta} \exp\left(\frac{[r(x) - \mathrm{VaR}_\beta(p^{\pi_k})]_+}{\eta(1-\beta)}\right). \tag{20}$$

The detailed derivation of equation 20 is provided in App. D.2.1. The update equation 20 acts as a *prior-anchored exponential tilt*. The first term anchors the distribution to a mixture of pre-trained prior and the last distribution to ensure stability, while the second term selectively amplifies the probability mass of samples exceeding the current Value-at-Risk. A visualization of the distribution updates is provided in Fig. 2. The iterative update equation 20 defines a distribution operator

$$(\mathcal{T}p)(x) \propto \left(p(x)\right)^{1-\alpha/\eta} \left(p^{\mathrm{pre}}(x)\right)^{\alpha/\eta} \exp\left(\frac{[r(x) - \mathrm{VaR}_\beta(p)]_+}{\eta(1-\beta)}\right). \tag{21}$$

In what follows, we show that our characterized target distribution is a fixed-point solution of this distribution operator.

**Proposition D.1** *Let $\beta \in (0, 1)$, $\alpha > 0$, $\eta > 0$. The characterized optimal distribution $p_R^\star$ in Theorem 4.1 is a fixed-point solution of the distribution operator defined in equation 21, i.e., $\mathcal{T}(p_R^\star) = p_R^\star$.*

Proposition D.1 confirms that TFFT effectively jumps directly to the convergence point of the FDC trajectory. By identifying the optimal threshold $t^*$ via offline data, TFFT bypasses the expensive iterative loop entirely, reducing the cost from $K$ fine-tuning runs to a single one.

## D.2  THEORY

### D.2.1  DERIVATION OF EQUATION 20

**Setup.**  Recalling that $\mathcal{G}(p) := \text{R-CVaR}_{\beta,p}[r] - \alpha D_{\mathrm{KL}}(p||p^{\mathrm{pre}})$, we have

$$\delta\mathcal{G}(p^k) = \frac{1}{1-\beta}\left[r(x) - \mathrm{VaR}_\beta(p^k)\right]_+ - \alpha\frac{\delta}{\delta p(x)} D_{\mathrm{KL}}(p^k||p^{\mathrm{pre}})$$

Recalling that $D_{\mathrm{KL}}(p||p^{\mathrm{pre}}) = \int p(x)\log\frac{p(x)}{p^{\mathrm{pre}}(x)}\,dx$, the functional derivative (first variation) is

$$\frac{\delta}{\delta p(x)} D_{\mathrm{KL}}(p||p^{\mathrm{pre}}) = \log\frac{p(x)}{p^{\mathrm{pre}}(x)} + 1. \tag{22}$$

Therefore, evaluated at $p^k$, we have

$$\frac{\delta}{\delta p(x)} D_{\mathrm{KL}}(p^k||p^{\mathrm{pre}}) = \log\frac{p^k(x)}{p^{\mathrm{pre}}(x)} + 1. \tag{23}$$

Then, we have

$$\delta\mathcal{G}(p^k) = \frac{1}{1-\beta}\left[r(x) - \mathrm{VaR}_\beta(p^k)\right]_+ - \alpha\left(\log\frac{p^k(x)}{p^{\mathrm{pre}}(x)} + 1\right) := f_k \tag{24}$$

At iteration $k$, the linearized subproblem has the generic form

$$p^{k+1} = \arg\max_p \langle f_k, p \rangle - \eta D_{\mathrm{KL}}(p||p^k), \tag{25}$$

**Closed-form solution of the KL-regularized linear GO.** We solve equation 25 by introducing the Lagrangian

$$\mathcal{L}(p, \nu) = \int p(x) f_k(x) \, dx - \eta \int p(x) \log \frac{p(x)}{p^k(x)} \, dx + \nu \Big( \int p(x) \, dx - 1 \Big).$$

Stationarity w.r.t. $p(x)$ gives

$$0 = \frac{\delta \mathcal{L}}{\delta p(x)} = f_k(x) - \eta \Big( \log \frac{p(x)}{p^k(x)} + 1 \Big) + \nu.$$

Rearranging and exponentiating yields the exponential-tilt form

$$p^{k+1}(x) = \frac{1}{Z_k} \, p^k(x) \exp\Big( \frac{f_k(x)}{\eta} \Big), \qquad Z_k = \int p^k(x) \exp\Big( \frac{f_k(x)}{\eta} \Big) \, dx. \qquad (26)$$

Thus, *even when $f_k$ contains the KL first variation at $p^k$, the subproblem still admits a closed-form solution because the optimization variable $p$ only appears inside $\langle f_k, p \rangle$ and the regularizer $D_{\mathrm{KL}}(p \| p^k)$.*

**Making the $p^k$-dependence explicit.** Substituting equation 24 into equation 26, we have

$$p^{k+1}(x) \propto p^k(x) \, \exp\Big( \frac{[r(x) - \mathrm{VaR}_\beta(p^k)]_+}{\eta(1-\beta)} \Big) \exp\Big( -\frac{\alpha}{\eta} \log \frac{p^k(x)}{p^{\mathrm{pre}}(x)} \Big) \exp\Big( -\frac{\alpha}{\eta} \Big)$$

$$\propto p^k(x) \, \exp\Big( \frac{[r(x) - \mathrm{VaR}_\beta(p^k)]_+}{\eta(1-\beta)} \Big) \Big( \frac{p^k(x)}{p^{\mathrm{pre}}(x)} \Big)^{-\alpha/\eta},$$

$$\propto \big( p^k(x) \big)^{1-\alpha/\eta} \big( p^{\mathrm{pre}}(x) \big)^{\alpha/\eta} \, \exp\Big( \frac{[r(x) - \mathrm{VaR}_\beta(p^k)]_+}{\eta(1-\beta)} \Big). \qquad (27)$$

The derivation is finished.

### D.2.2 PROOF OF PROPOSITION D.1

For simplicity of notation, write

$$s^\star(x) \triangleq \frac{[r(x) - t^\star]_+}{1-\beta}.$$

Then, equation 6 can be rewritten as

$$p_R^\star(x) = \frac{1}{Z(t^\star)} p^{\mathrm{pre}}(x) \exp\Big( \frac{s^\star(x)}{\alpha} \Big). \qquad (28)$$

Since $\mathrm{VaR}_\beta(p_R^\star) = t^\star$, we have

$$\frac{[r(x) - \mathrm{VaR}_\beta(p_R^\star)]_+}{\eta(1-\beta)} = \frac{[r(x) - t^\star]_+}{\eta(1-\beta)} = \frac{s^\star(x)}{\eta}.$$

Recall the definition of $\mathcal{T}$ in equation 21. Applying $\mathcal{T}$ to $p_R^\star$ yields

$$(\mathcal{T} p_R^\star)(x) \propto \big( p_R^\star(x) \big)^{1-\alpha/\eta} \big( p^{\mathrm{pre}}(x) \big)^{\alpha/\eta} \, \exp\Big( \frac{s^\star(x)}{\eta} \Big). \qquad (29)$$

Substitute equation 28 into $\big( p_R^\star(x) \big)^{1-\alpha/\eta}$ yields

$$\big( p_R^\star(x) \big)^{1-\alpha/\eta} = \Big( \frac{1}{Z(t^\star)} p^{\mathrm{pre}}(x) e^{s^\star(x)/\alpha} \Big)^{1-\alpha/\eta} = Z(t^\star)^{-(1-\alpha/\eta)} \big( p^{\mathrm{pre}}(x) \big)^{1-\alpha/\eta} \exp\Big( \frac{1-\alpha/\eta}{\alpha} s^\star(x) \Big).$$

Plugging this into equation 29 yields

$$(\mathcal{T} p_R^\star)(x) \propto Z(t^\star)^{-(1-\alpha/\eta)} \big( p^{\mathrm{pre}}(x) \big)^{1-\alpha/\eta} \big( p^{\mathrm{pre}}(x) \big)^{\alpha/\eta} \, \exp\Big( \frac{1-\alpha/\eta}{\alpha} s^\star(x) \Big) \exp\Big( \frac{s^\star(x)}{\eta} \Big).$$

Since $\frac{1-\alpha/\eta}{\alpha} + \frac{1}{\eta} = \frac{1}{\alpha} - \frac{1}{\eta} + \frac{1}{\eta} = \frac{1}{\alpha}$, we have

$$(\mathcal{T} p_R^\star)(x) \propto p^{\mathrm{pre}}(x) \exp\Big( \frac{s^\star(x)}{\alpha} \Big) \propto p_R^\star(x),$$

where the proportionality constants are absorbed by normalization. Hence $\mathcal{T}(p_R^\star) = p_R^\star$. The proof is complete.

## D.3 SETUP AND IMPLEMENTATION DETAILS

**Distribution-level FDC operator.** To separate distribution dynamics from model approximation effects, we study FDC directly in distribution space on a discretized 2D domain. Starting from $p^0 = p^{\mathrm{pre}}$, the iterates are defined by

$$p^{k+1}(x) \;\propto\; \big(p^k(x)\big)^{1-\alpha/\eta_k} \big(p^{\mathrm{pre}}(x)\big)^{\alpha/\eta_k} \exp\Big(\frac{[r(x) - \mathrm{VaR}_\beta(p^k)]_+}{\eta_k(1-\beta)}\Big), \tag{30}$$

where $\mathrm{VaR}_\beta(p^k)$ denotes the $\beta$-quantile of the random variable $r(X)$ under $X \sim p^k$, and $\{\eta_k\}_{k=1}^K$ is an increasing schedule.

**Step-size parameter** $\eta_k$ The closed-form update equation 20 provides guidance for choosing $\eta$, which can be interpreted as an inverse step size in the sequence of linearized GO subproblems. In practice, we enforce $\alpha/\eta < 1$ so the exponent on $p^{\pi_k}$ remains positive, yielding stable distribution dynamics. Moreover, $\alpha/\eta$ controls the strength of anchoring: early in training, when $p^{\pi_k}$ may be far from optimal, a larger $\alpha/\eta$ (stronger pull toward $p^{\mathrm{pre}}$) is often beneficial for stability. Later, once $p^{\pi_k}$ approaches the target distribution, the prior can become overly restrictive; then a smaller $\alpha/\eta$ is preferred to let the tilt term dominate, which corresponds to using a larger $\eta$ (i.e., a smaller effective step size $1/\eta$).

**Closed-form target distribution.** We compare $\{p^k\}$ against the closed-form target distribution

$$p^\star(x) \;\propto\; p^{\mathrm{pre}}(x) \exp\Big(\frac{[r(x) - t^\star]_+}{\alpha(1-\beta)}\Big),$$

where $t^\star$ is the minimizer in Theorem 4.1, i.e.,

$$t^\star \in \arg\min_{t\in\mathbb{R}} \left\{ t + \alpha \log \mathbb{E}_{X\sim p^{\mathrm{pre}}}\left[\exp\Big(\frac{[r(X) - t]_+}{\alpha(1-\beta)}\Big)\right] \right\}.$$

We compute $t^\star$ via one-dimensional numerical minimization (golden-section search) and then construct $p^\star$ by normalization on the grid.

**2D prior and reward.** We use a standard Gaussian prior $p^{\mathrm{pre}} = \mathcal{N}(0, I_2)$. The reward is chosen as $r(x) = \exp\big(-\frac{1}{2}\|x - \mu\|^2/\sigma^2\big)$ with center $\mu = (2, 2)$ and scale $\sigma = 0.8$. We select $\alpha = 1$, $\beta = 0.9$.

**Discretization and numerical computation.** We discretize $\mathcal{X} = [-4, 4] \times [-4, 4]$ using an $n \times n$ uniform grid and represent each distribution as a probability mass function (PMF) on grid cells. Densities shown in figures are obtained by dividing the PMF by the cell area. To compute $\mathrm{VaR}_\beta(p^k)$, we form the weighted empirical distribution of $\{r(x_i)\}$ with weights $p^k(x_i)$ and evaluate the weighted $\beta$-quantile using linear interpolation. All updates in equation 30 are implemented in log-space followed by normalization.

**Evaluation metrics.** We report information distances between iterates and the target distribution, including $\mathrm{KL}(p^k\|p^\star)$, Jensen–Shannon divergence $\mathrm{JS}(p^k, p^\star)$, and total variation distance

$$\mathrm{TV}(p^k, p^\star) = \frac{1}{2}\int_{\mathcal{X}} |p^k(x) - p^\star(x)|\, dx,$$

approximated by summation on the grid. We also track the threshold sequence $t_k = \mathrm{VaR}_\beta(p^k)$ and compare it to $t^\star$.

## D.4 ADDITIONAL FIGURES ON FDC

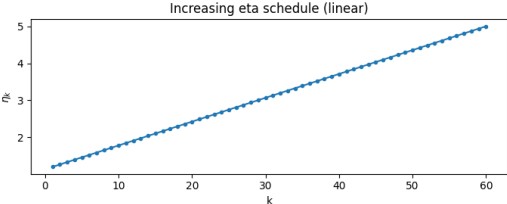

Figure 6: Increasing $\eta$ schedule used in FDC experiments. We use a time-varying regularization parameter $\{\eta_k\}_{k=1}^{K}$ that increases with iteration $k$. This annealing stabilizes the later-stage dynamics of the FDC distribution operator.

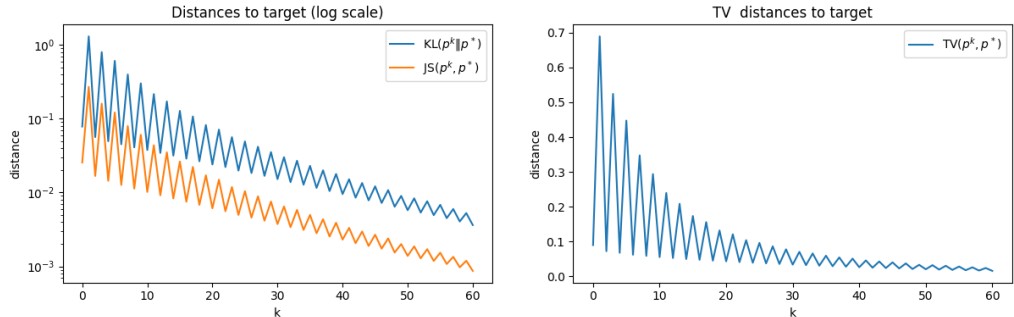

Figure 7: Distance to the target distribution. We report distances between $p^k$ and $p^*$, including KL divergence, JS divergence, and total variation distance. All of the distances converge to 0, indicating that the distribution converges to the target.

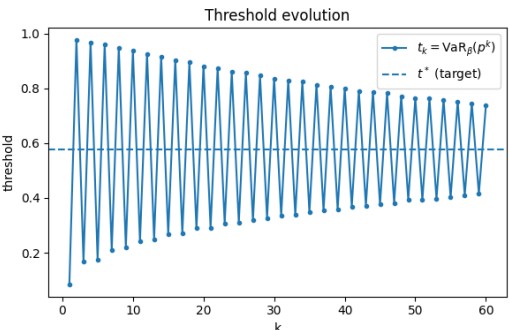

Figure 8: Threshold evolution under FDC. We observe that the sequence $t_k = \text{VaR}_\beta(p^k)$ induced by the FDC updates converges to the optimal threshold $t^\star$ returned by the dual minimization in Theorem 4.1. This provides an additional evidence for the convergence of the distributions.

# E   EXPERIMENTAL DETAILS

Most of the experiments were conducted on computing clusters equipped with NVIDIA A100 and A100 (80GB) GPUs. For all illustrative experiments we utilize Adjoint Matching (AM) Domingo-Enrich et al. (2024) for the entropy-regularized fine-tuning solver.

## E.1   2D EXAMPLE

### E.1.1   IMPLEMENTATION

For the 2D illustrative analysis, we directly compute the optimal distributions via importance sampling. We define the pre-trained prior distribution $p^{\text{pre}}$ as a standard 2D Gaussian $\mathcal{N}(0, I_2)$. The reward function is defined as the linear sum of coordinates $r(x) = x_1 + x_2$. We set the temperature parameter $\alpha = 1.0$. We draw $N = 10000$ samples from the prior to serve as an offline dataset to compute the optimal $t$ in the first stage. The probability density functions (PDFs) are estimated from the weighted samples using Gaussian Kernel Density Estimation with a bandwidth factor of 0.25.

### E.1.2   ADDITIONAL RESULTS IN THE 2D EXAMPLE

Table 4: Results on 2D example. Expected fine-tuning (EXP-FT) achieves highest expected rewards, right-CVaR fine-tuning (R-TFFT) achieves the highest right-CVaR value, and left-CVaR fine-tuning (L-TFFT) achieves highest left-CVaR value.

| Model | $E[r]$ | R-CVaR$_{0.8}[r]$ | L-CVaR$_{0.2}[r]$ |
|---|---|---|---|
| Pre-trained | 0.00 | 1.99 | -1.98 |
| EXP-FT | **2.03** | 4.06 | 0.03 |
| L-TFFT | 2.00 | 3.08 | **1.23** |
| R-TFFT | 1.26 | **6.31** | -1.80 |

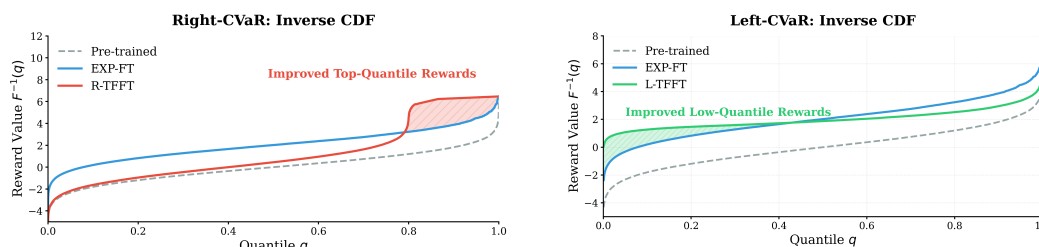

Figure 9: Inverse CDFs of the pre-trained model, expected fine-tuning model, left- and right-CVaR fine-tuning model in the 2D example. R-TFFT achieves higher rewards in the high-quantile region while L-TFFT achieves higher rewards in the low-quantile region.

## E.2   TEXT-TO-IMAGE GENERATION

### E.2.1   IMPLEMENTATION

**Experimental setup:**   Following Domingo-Enrich et al. (2024), we utilize Stable Diffusion v1-5 Rombach et al. (2022) as the pre-trained generative backbone. To align the model with human aesthetic preferences, we employ ImageReward Xu et al. (2023) as the downstream reward function. We set $\alpha = 1$ and the reward function

$$r(x) = 100 \times \text{ImageReward}(x).$$

**Training details:**   The training follows the two-stage procedure proposed in Algorithm 1. In stage 1, we estimate the optimal threshold $t^*$ by maximizing the concave objective $F_L(t)$. We collect an offline batch of 10000 samples generated from the pre-trained prior $p^{\text{pre}}$. With $\beta = 0.2$, the resulting optimal threshold was found to be approximately $t^* \approx 0.700759$. In stage 2, We perform a single

fine-tuning run using the Adjoint Matching solver with the pseudo-reward $r^*(x) = -[t^* - r(x)]_+/\beta$. Both EXP-FT and L-TFFT are trained for 1600 steps with a step size of $3e^{-6}$. Since the pseudo-reward transformation incurs negligible overhead, EXP-FT and L-TFFT exhibit nearly identical training times ($\sim$82 hours on a single NVIDIA A100 80GB GPU). To ensure a fair comparison under a fixed budget, we constrain FDC to the same total duration by running 2 outer iterations ($K = 2$) with 800 steps per iteration.

**Evaluation metrics:**   We evaluate the performance using four complementary metrics to ensure a comprehensive assessment. e report the expected ImageReward to measure general human preference alignment. To assess reliability, we report the **left-CVaR** metric (average of the bottom $\beta = 20\%$ rewards), which quantifies the model's performance in worst-case scenarios. We compute the CLIP Score Hessel et al. (2021) to evaluate the semantic consistency between the generated images and the input text prompts. Human Preference Score v2 (HPSv2) Wu et al. (2023) is used as a robust proxy for human evaluation to test the model's domain generalization capabilities beyond the training reward model. To measure the diversity of the generated outputs, we compute the DreamSim score Fu et al. (2023), which assesses perceptual similarity.

### E.2.2   ADDITIONAL RESULTS IN IMAGE GENERATION

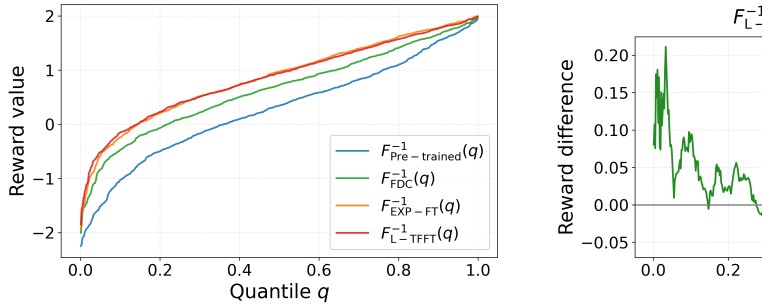

Figure 10: Comparison of reward distributions on stable-diffusion. The left panel shows the inverse CDF of all four methods, while the right panel illustrates the difference of inverse CDFs between the expected and CVaR fine-tuning, which clearly shows that L-TFFT outperforms EXP-FT in lower tail $q \in [0, 0.2]$.

Fig. 10 plots the inverse Cumulative Distribution Functions (CDFs) for all the methods. We observe in the left panel that EXP-FT and L-TFFT are leading in most quantile values, indicating their significantly improved rewards. In the quantile region $q \in [0, 0.2]$, L-TFFT outperforms EXP-FT, i.e., $F_{\text{L-TFFT}}^{-1}(q) > F_{\text{EXP-FT}}^{-1}(q)$. The difference is amplified in the right panel, where we observe a clear positive gap, confirming that L-TFFT successfully raises the quality floor of worst-case samples.

Fig. 11 visualizes the empirical PDFs for the Pre-trained, EXP-FT, and L-TFFT models, with the dashed vertical line indicating the $20\%$ quantile value. As shown in the top panel, the Pre-trained model exhibits a broad distribution with a heavy left tail, indicating a high frequency of low-reward (low-quality) samples. The EXP-FT method successfully shifts the entire distribution to the right, but the distribution retains a non-negligible density in the negative reward region. In contrast, L-TFFT further suppresses the density in the extreme lower tail, as evidenced by a higher 20% quantile value.

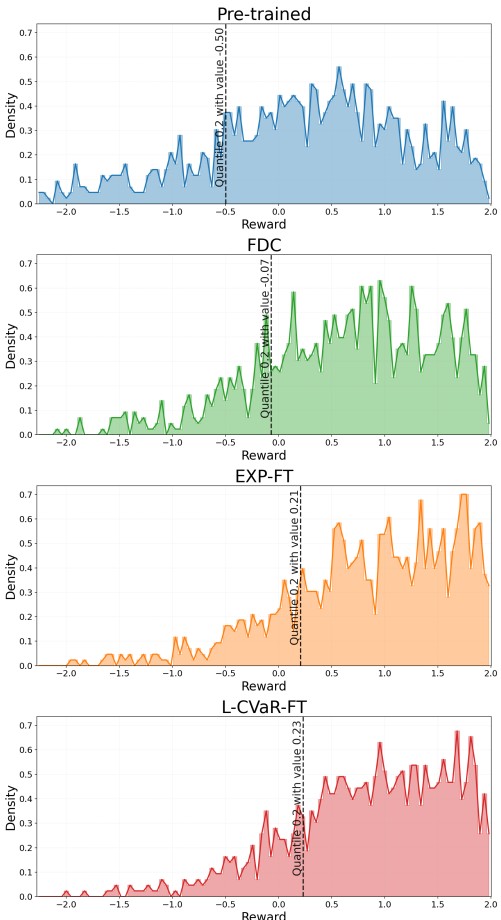

Figure 11: Reward PDFs for Pre-trained, FDC, EXP-FT, and L-TFFT models in text-to-image generation. The vertical dashed line represents the 0.2-quantile. L-TFFT effectively suppresses the probability mass in the lower tail (worst-case scenarios) compared to the baselines, which is evidenced by a higher 20% quantile value.

### E.3   MOLECULE DESIGN

#### E.3.1   IMPLEMENTATION

**Experimental setup:**   We employ FlowMol Dunn and Koes (2025) as the backbone generative flow model, which is pre-trained on the GEOM-Drugs dataset Axelrod and Gomez-Bombarelli (2022) to generate 3D molecular conformations. To optimize for chemical stability, we define the reward function $r(x)$ as the negative energy calculated using GFN1-xTB Friede et al. (2024), a semi-empirical quantum mechanical method. Lower energy corresponds to higher stability, so we maximize the negative energy. We set $\alpha = 1$ and the reward function

$$r(x) = 6 \times (-\text{Energy}(x)).$$

**Training details:**   The training utilizes our TFFT framework, specifically configured for the right-CVaR objective to discover novel samples. In Stage 1, we minimize the convex objective $F_R(t)$ to estimate the target threshold $t^*$ corresponding to the top 10% of the distribution ($\beta = 0.9$). We approximate the expectations using a fixed offline batch of 10000 molecules sampled from the pre-trained model. In **Stage 2**, we fine-tune the model using the Adjoint Matching solver. Directly targeting the top 10% ($\beta = 0.9$) creates a sparse signal, as few samples initially exceed the threshold. To accelerate training, we employ a $\beta$-annealing schedule: we initialize the quantile at $\beta_0 = 0$

(effectively standard expected fine-tuning) and gradually increase it to the target $\beta = 0.9$. This curriculum allows the model to improve the general distribution before focusing on the extreme tail. The model is trained for a total of 120 gradient steps using the AdamW optimizer with a learning rate of 1e-4 and batch size 8.

**Evaluation metrics:** We evaluate performance by generating 2,000 samples per model across three independent runs. The evaluation procedure is as follows. First, we verify validity for all generated molecules, determined by whether molecules can pass RDKit sanitization checks. Second, we compute the Average Reward and Synthetic Accessibility (SA) Score exclusively on the subset of valid molecules. This filtration step is critical because invalid structures can occasionally exhibit anomalously low energy values (high rewards) due to simulation artifacts.

### E.3.2 ADDITIONAL RESULTS IN MOLECULE DESIGN

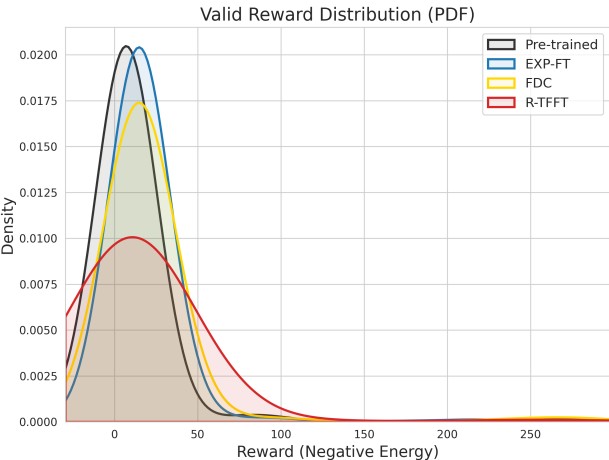

Figure 12: PDF of valid rewards of different methods in molecule design.

To rigorously assess the quality of the generated structures, we analyze the reward distributions exclusively for the valid molecules, i.e., those passing RDKit sanitization. As shown in Figure 12, all fine-tuning methods successfully shift the reward distribution to the right compared to the Pre-trained baseline. Notably, **R-TFFT** exhibits the highest density in the high-reward interval of $[50, 100]$, confirming its ability to concentrate probability mass on chemically stable configurations.

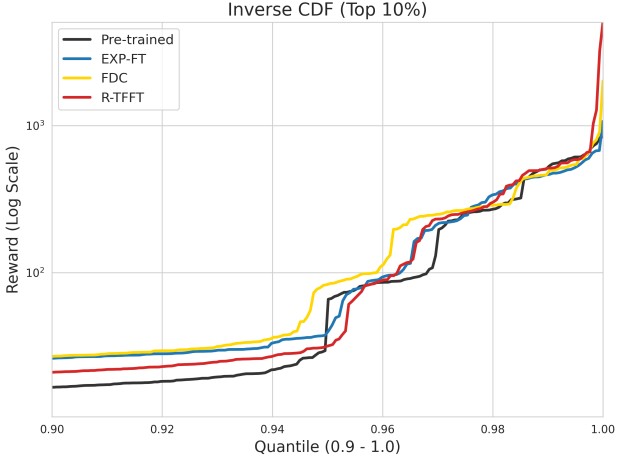

Figure 13: Inverse CDF of valid rewards of different methods in molecule design.

Figure 13 analyzes the top 10% of the data ($q \in [0.9, 1.0]$), uncovering a notable performance shift. In the intermediate range ($q \approx 0.94 - 0.99$), dominance alternates among methods, with sometimes even the Pre-trained model occasionally outperforming R-TFFT. We attribute this to a trade-off between aggressive reward optimization and chemical validity; fine-tuning methods may generate high-energy but invalid structures that are subsequently filtered, leaving a valid subset that temporarily lags in reward compared to the more consistently valid Pre-trained baseline. Crucially, as $q \to 1$, our R-TFFT re-establishes dominance in the extreme tail. This confirms that despite validity constraints, the right-CVaR objective successfully identifies valid and novel candidates.

### E.4 ABLATION STUDY ON THE RISK LEVEL $\beta$

We study the effect of the risk level $\beta$ in the molecule design task. Larger $\beta$ focuses on a smaller novel portion of the high-reward tail, which makes optimization harder and tends to produce more invalid samples. Table 5 reports the validity rate across 5 runs: a moderate choice ($\beta = 0.9$) achieves the highest validity with the lowest variance, whereas more aggressive tail focusing ($\beta = 0.95, 0.99$) reduces validity and increases variability.

Table 5: Ablation on the risk level $\beta$ for molecule design. We report validity rate (%) mean and standard deviation across 5 runs.

| $\beta$ | 0.8 | 0.9 | 0.95 | 0.99 |
|---|---|---|---|---|
| Validate rate | $78.1\% \pm 13.9\%$ | $81.2\% \pm 3.9\%$ | $69.4\% \pm 21.5\%$ | $61.5\% \pm 27.6\%$ |

### E.5 SENSITIVITY TO THRESHOLD OPTIMIZATION

To validate the theoretical bounds proposed in Theorem 6.2, we conducted numerical experiments on a synthetic 2D Gaussian task where the optimal target distribution $p^*$ can be computed exactly. We use right-CVaR as an example. Recall that Theorem 6.2 defines the scaling constant as $\lambda = \alpha(1 - \beta)$. We investigate the impact of the coefficient $\alpha$ and the quantile $\beta$.

We define the base distribution $p^{pre}(x)$ as a standard Gaussian $\mathcal{N}(0, I)$. We introduce an estimation error $\delta = |\hat{t} - t^*|$ to the threshold and compute the KL divergence between the ideal target distribution $p^*(\cdot; t^*)$ and the approximate distribution $\hat{p}(\cdot; \hat{t})$. Fig. 14 shows how the KL divergence increases with $\delta$ with different $\alpha$ and $\beta$. As expected, larger $\delta$ monotonically increases the KL divergence error. Moreover, the degradation is more pronounced when using a larger quantile $\beta$ and a smaller penalty coefficient $\alpha$.

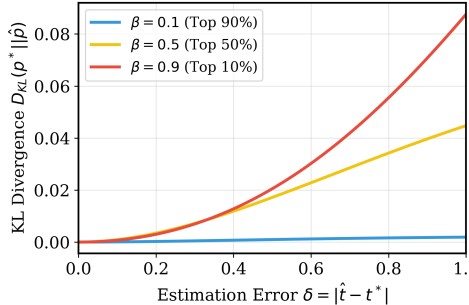 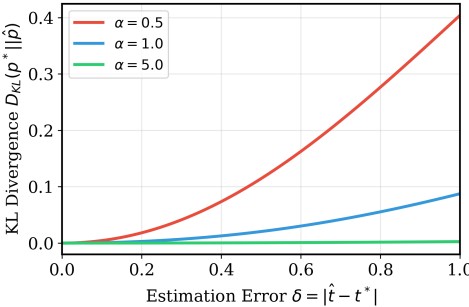

Figure 14: Sensitivity to threshold optimization with different values of $\alpha$ and $\beta$.

## F  ADDITIONAL TEXT-TO-IMAGE SAMPLES

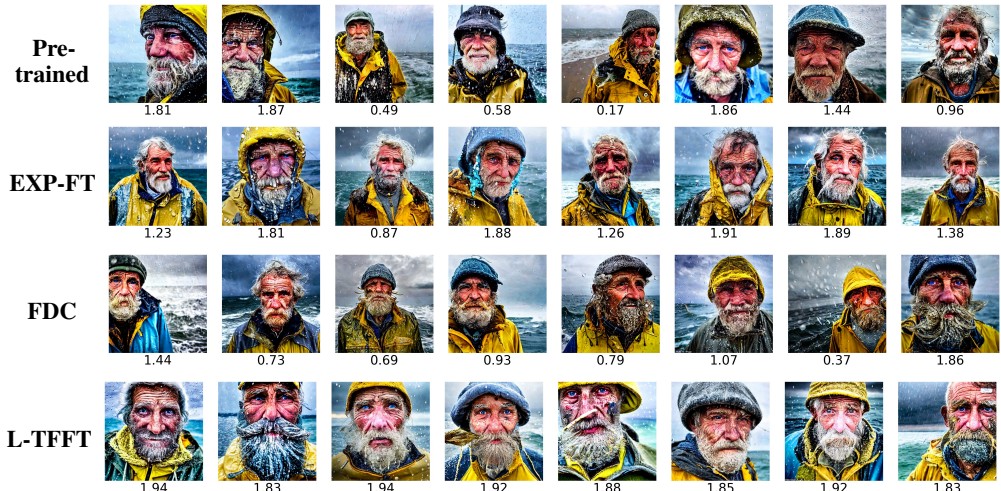

Figure 15: Generated samples with ImageReward values. Prompt: "A documentary photograph of an elderly fisherman with a weather-beaten face and bushy grey beard. His piercing **blue** eyes look out at a stormy sea. He is wearing a heavy, waterproof, dirty yellow oilskin jacket covered in salt spray and grime. Raindrops on his face". Images with low rewards fail to render blue eyes.

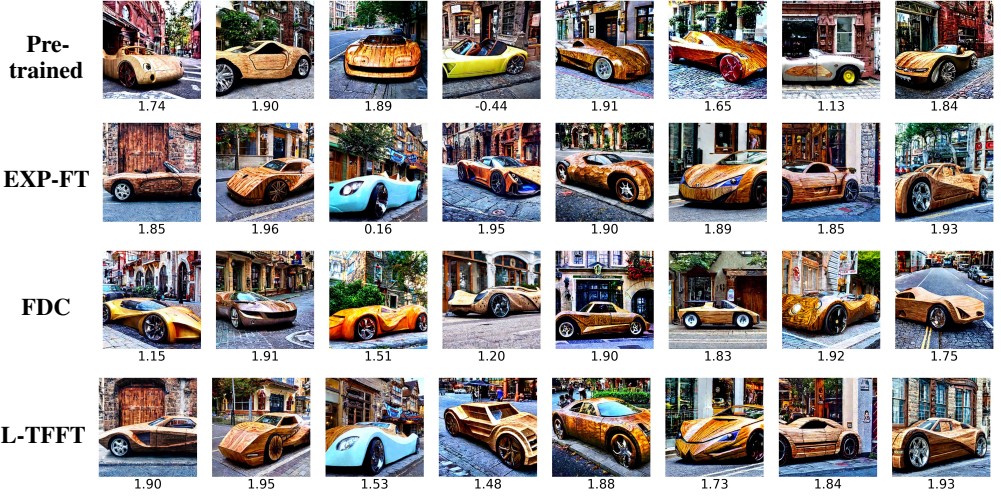

Figure 16: Generated samples from different models with ImageReward values. Prompt: "A sports car carved entirely out of polished wood, parked on a street".

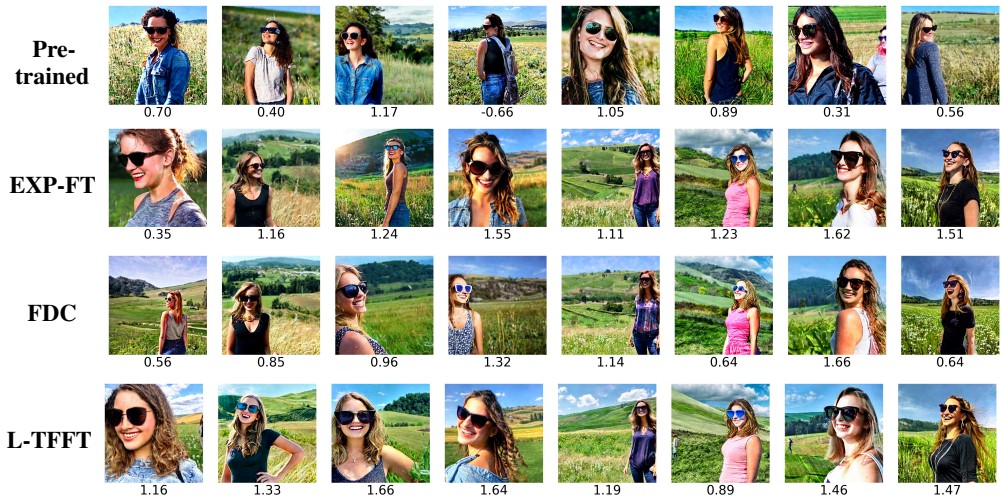

Figure 17: Generated images with ImageReward values from different models. Prompt: "The beautiful young woman in sunglasses is standing at the background of field and hill. She is smiling and looking over shoulder".

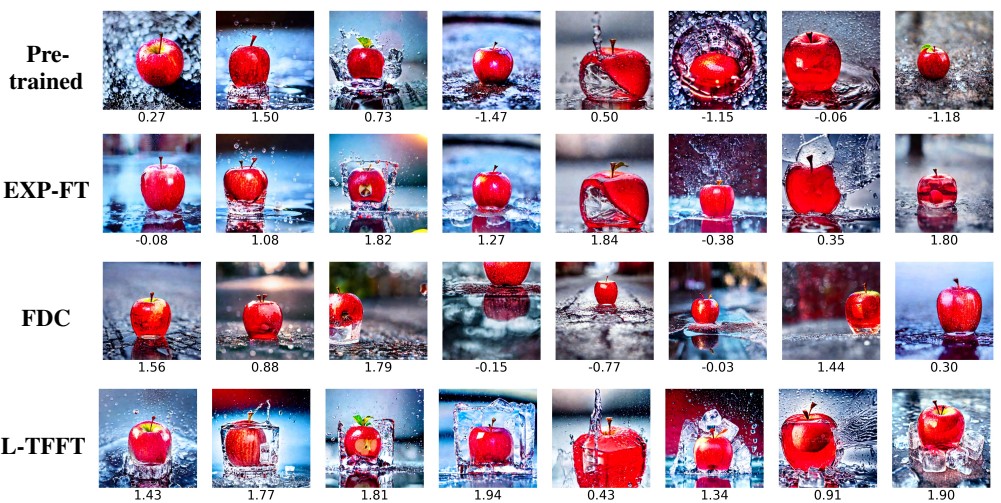

Figure 18: Generated samples from different models with ImageReward values. Prompt: "A red apple inside a clear ice cube, the ice cube is melting on a hot pavement, macro photography".

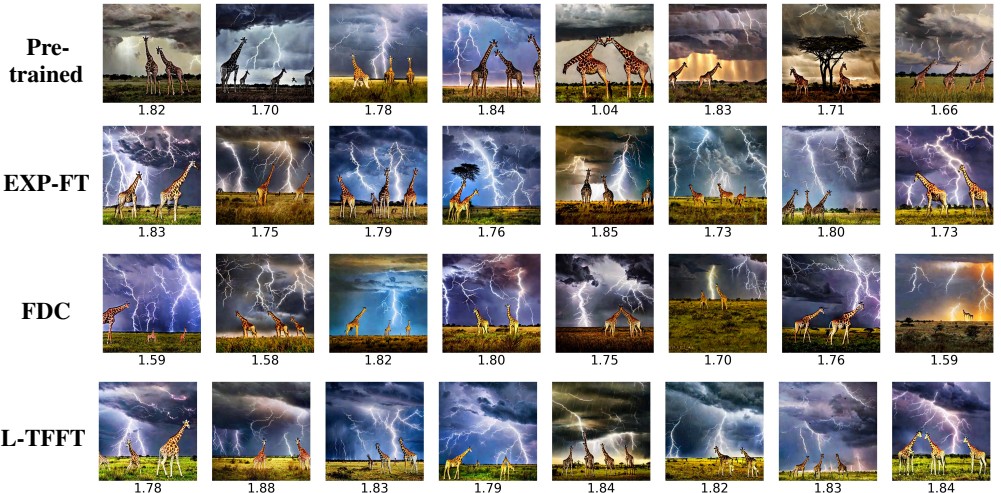

Figure 19: Generated samples from different models with ImageReward values. Prompt: "Award-winning wildlife photograph of giraffes standing on a savannah under distant lightning, dramatic storm clouds, rain curtains in the background, warm foreground light, cinematic contrast, sharp detail on coats and grass".

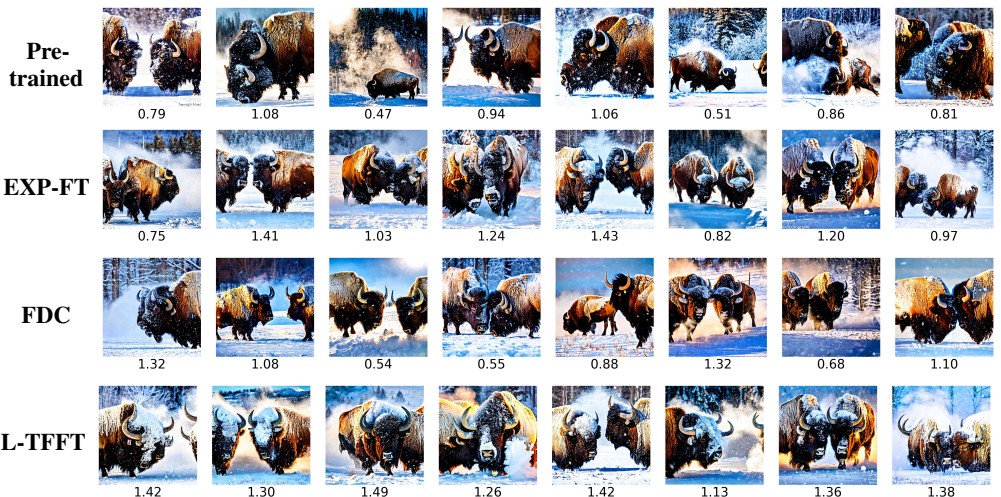

Figure 20: Generated samples from different models with ImageReward values. Prompt: "National Geographic style photography of two massive American bison bulls colliding head-on in deep winter snow. A massive explosion of powder snow is kicked up and frozen around them, obscuring parts of their bodies. Their breath forms thick clouds of steam in the freezing air. Low, cold winter sun creating long shadows. Telephoto lens compression".

