# OpenReview forum: "Efficient Tail-Aware Generative Optimization via Flow Model Fine-Tuning"
_ICLR.cc/2026/Workshop/FM4Science — ICLR 2026 Workshop FM4Science Poster_

### Official Review · Reviewer_rPbL · 2026-02-22
**Efficient Tail-Aware Generative Optimization via Two-Stage CVaR Fine-Tuning**

**Rating:** 7
**Confidence:** 3

**Review:**

**Summary**

This paper studies reward-based fine-tuning of generative models from a *tail-risk* perspective. Instead of optimizing only expected reward, it targets the distribution tails: the left tail (avoiding low-reward failures for reliability/safety) and the right tail (increasing probability of rare high-reward samples for discovery). The proposed method, **TFFT (Tail-aware Flow Fine-Tuning)**, uses a variational dual form of CVaR to decompose optimization into (i) a lightweight 1D threshold search for $t^*$ using samples from the pretrained model, and (ii) a single entropy-regularized fine-tuning run using a derived pseudo-reward. The key claim is that this achieves tail control with computational cost close to standard expected-reward fine-tuning, while avoiding multi-round solver calls required by some baselines.

**Strengths**

1. Clear and practical motivation: tail behavior (worst-case failures or rare best cases) is often more relevant than mean reward.
2. Simple, clean algorithm: CVaR optimization is reduced to “threshold search + one standard entropy-regularized fine-tune”, making it easy to implement and plug into existing solvers.
3. Helpful theory: the threshold objective has nice optimization properties (convex/concave and smoothness), and the paper provides a sensitivity bound relating threshold error to the final KL deviation.
4. Empirical evidence of tail shaping: toy and higher-dimensional experiments (text-to-image and molecules) show improvements on tail metrics (e.g., L-CVaR / R-CVaR), and report reduced training cost relative to multi-call approaches.

**Weaknesses**

1. Budget accounting could be clearer. While Stage 1 is described as lightweight, in practice the dominant cost may be reward evaluation (e.g., ImageReward or molecular scoring). A more explicit wall-clock breakdown (Stage 1 sampling/reward eval vs Stage 2 training) and strict budget matching would strengthen the “efficient” claim.
2. Assumptions on reward distribution. Some theoretical arguments rely on continuity/no-atoms and boundedness-type assumptions; real rewards can be discrete, clipped, or have many ties, which may affect VaR/threshold stability. Discussion (or small diagnostics) on such cases would improve robustness.
3. Trade-offs are not fully characterized. Tail improvement may come with diversity/mean trade-offs (e.g., reduced diversity).
4. Baseline coverage. The main comparisons are against expected-reward fine-tuning and a multi-call baseline (e.g., FDC). Adding more risk-sensitive or distributional baselines (even simplified) would strengthen the empirical positioning.

---

### Official Review · Reviewer_ZMG3 · 2026-02-23
**Efficient Tail-Aware Generative Optimization via Flow Model Fine-Tuning**

**Rating:** 9
**Confidence:** 3

**Review:**

Summary:

This paper introduces Tail-aware Flow Fine-Tuning (TFFT), a CVaR-based fine-tuning method for diffusion/flow generative models that explicitly shapes reward tails instead of only optimizing the mean reward. The main technical idea is a variational dual reformulation of right-/left-CVaR that decomposes the nonlinear objective into (1) a cheap 1D threshold search using samples from the pretrained model, and (2) a single entropy-regularized fine-tuning run using a threshold-defined pseudo-reward. The paper evaluates (i) toy 2D settings, (ii) risk-averse text-to-image tuning on Stable Diffusion with ImageReward (improves lower-tail CVaR under a fixed GPU budget), and (iii) novelty-seeking molecular design with FlowMol (improves upper-tail CVaR with fewer fine-tuning calls than an iterative baseline).

Pros:
1. Clear motivation: many real deployments care about worst-case failures (left tail) or rare best-case discoveries (right tail), which expected-reward fine-tuning does not target directly.
2. The dual CVaR decomposition is clean and practically appealing: offline scalar threshold optimization + one standard fine-tuning run via a pseudo-reward.
3. Strong empirical signal on the stated goals: on text-to-image, L-TFFT matches mean ImageReward while improving L-CVaR (quality floor) under the same compute; on molecules, R-TFFT improves R-CVaR and finds higher-reward top candidates while keeping validity relatively high.
4. Useful theory beyond “just works”: convex/concave properties for the threshold objective and a stability bound connecting threshold error to distributional deviation.

Cons:
1. Experimental coverage of hyperparameters is thin: tail behavior depends strongly on β (risk level) and α (KL strength), but sensitivity/robustness across β/α is not thoroughly explored beyond a couple fixed settings.
2. The method assumes the pretrained sampler can provide enough informative samples for threshold estimation; in high-dimensional domains where tail events are extremely rare, Stage-1 estimation may be sample-hungry in practice.
3. The main baselines are expected fine-tuning and FDC; broader comparisons to other distributional/risk-sensitive tuning methods (or alternative tail objectives beyond CVaR) would strengthen positioning.

---

### Decision · Program_Chairs · 2026-03-03

Accept (Poster)